



# High-resolution InSAR Regional Soil Water Storage Above Permafrost

Yue Wu[1], Jingyi Chen[1,2,3], M. Bayani Cardenas[3], and George W. Kling[4]

[1]University of Texas at Austin, Aerospace Engineering and Engineering Mechanics, Texas, United States
[2]University of Texas at Austin, Center of Space Research, Texas, United States
[3]University of Texas at Austin, Earth and Planetary Sciences, Texas, United States
[4]University of Michigan, Ecology and Evolutionary Biology, Michigan, United States

**Correspondence:** Yue Wu (sophywu@utexas.edu)

**Abstract.** The hydrology of thawing permafrost affects the fate of the vast amount of permafrost carbon due to its controls on waterlogging, redox status, and transport. However, regional mapping of soil water storage in the soil layer that experiences the annual freeze-thaw cycle above permafrost, known as the active layer, remains a formidable challenge over remote arctic regions. This study shows that Interferometric Synthetic Aperture Radar (InSAR) observations can be used to estimate the amount of soil water originating from the active layer seasonal thaw. Our ALOS InSAR results, validated by in situ obser-
vations, show that the thickness of the soil water that experiences the annual freeze-thaw cycle ranges from 0 to 75 cm in a 60-by-100-km area near the Toolik Field Station on the North Slope of Alaska. Notably, the spatial distribution of the soil water correlates with surface topography and land vegetation cover types. We found that pixel-mismatching of the topographic map and radar images is the primary error source in the Toolik ALOS InSAR data. The amount of pixel misregistration, the local slope, and the InSAR perpendicular baseline influence the observed errors in InSAR Line-Of-Sight (LOS) distance measure-
ments non-linearly. For most of the study area with a percent slope of less than 5%, the LOS error from pixel misregistration is less than 1 cm, translating to less than 14 cm of error in the soil water estimates.

## 1 Introduction

Permafrost soils in the Arctic store twice the amount of carbon found in the atmosphere (Hugelius et al., 2014; Ping et al.,
2008). Over the past decades, warming has led to permafrost thawing (Jorgenson et al., 2006), which may result in the release of stored organic matter into the atmosphere as greenhouse gases and further amplify global warming (Serreze and Barry, 2011; Schaefer et al., 2014; Schuur et al., 2015). In permafrost regions, groundwater flows through the topmost portion of the soil, known as the active layer, that freezes and thaws annually (Woo, 2012; O'Connor et al., 2020). This groundwater flow contains carbon and is important in the export of carbon from land to the ocean and atmosphere (Kling et al., 1991; Stieglitz et al., 2003;
Walvoord and Striegl, 2007; Vonk and Gustafsson, 2013; Paytan et al., 2015; Neilson et al., 2018). To understand how thawing permafrost contributes to the global carbon cycle, it is important to understand the hydrologic flow and transport processes in the active layer. Whether the carbon held by the active layer soils will be transformed to carbon dioxide or methane (a more





powerful greenhouse gas), or whether it will flow towards rivers and lakes as dissolved carbon in groundwater, depends largely on the wetness or dryness of the active layer (i.e., how much water is stored).

Most of the arctic permafrost region is hard to access, and in situ observations of water storage and water flow in the active layer are extremely limited. Remote sensing techniques hold promise for local to regional observation of the hydrologic properties and hydrologic states of permafrost. For example, observations from the Gravity Recovery and Climate Experiment (GRACE) mission detect changes in permafrost water mass over a regional scale (Muskett and Romanovsky, 2009), but the spatial resolution is too coarse ($\sim$ 100s of km) to be used in most hydrologic models (Text S1). In comparison, by measuring

the phase difference between two paired radar images, Interferometric Synthetic Aperture Radar (InSAR) techniques estimate surface deformation between the two radar acquisition times along the radar Line-Of-Sight (LOS) direction (Rosen et al., 2000; Hanssen, 2001) at the spatial scale ($\sim$ 10s to 100s meters spatial resolution) that overlaps with the scale of hydrologic field measurements and modeling grids. Although spaceborne InSAR has been used for estimating surface deformation associated with solid earth processes since the 1990s (Massonnet et al., 1993; Fialko et al., 2002; Pritchard and Simons, 2002; Shirzaei

et al., 2013; Chen et al., 2014), it has only been recently used to estimate surface deformation associated with the seasonal freeze-thaw process of the soil active layer (Liu et al., 2010; Short et al., 2011; Antonova et al., 2018; Strozzi et al., 2018; Rouyet et al., 2019). Because ice density is less than water density (and thus ice volume is greater than water volume), the land surface subsides as the active layer thaws from winter to summer (Liu et al., 2010). Furthermore, InSAR-observed long-term subsidence trend signals over permafrost terrain have been used to study the deepening of the active layer due to wildfires or

excessive melt of ground ice (Michaelides et al., 2019; Liu et al., 2014, 2015; Iwahana et al., 2016; Yanagiya and Furuya, 2020; Abe et al., 2020; Eshqi Molan et al., 2018).

    Existing InSAR permafrost studies tended to associate the magnitude of the InSAR-observed thaw subsidence with the active layer thickness (Liu et al., 2012; Schaefer et al., 2015; Chen et al., 2021). However, the amplitude of the thaw subsidence and frost heave could depend on other factors such as sediment type and local topographic slope (Daout et al., 2017). Our recent

study found that the amplitude of the seasonal thaw subsidence signal is linearly proportional to the total amount of liquid water that experiences the ice-to-water phase change in a single thaw season (Chen et al., 2020). Therefore, satellite remote sensing of surface deformation is a potential strategy for mapping water storage in the active layer with broad coverage and relatively high spatial resolution. In this paper, our goal is to further advance and apply InSAR techniques for high-resolution above-permafrost water storage mapping. Here, we analyze ALOS PALSAR data over a much larger area of undisturbed

permafrost terrain in the Arctic Foothills to estimate the amount of active layer soil water from ground ice melting during summer thaw seasons (denoted as $z_{water}$). We evaluate the accuracy of the InSAR algorithm using in situ soil measurements collected at over 200 remote sites within $\sim$ 100 km around the Toolik Field Station as well as optical imagery and land cover maps. An important contribution of this work is the identification of the primary error sources in Toolik ALOS PALSAR Line-Of-Sight (LOS) measurements, which are errors in the DEM data and DEM-SAR pixel misregistration. While it is common to

assume that DEM errors in InSAR LOS measurements tend to occur in steep terrains and their magnitude is proportional to the perpendicular baseline, our results show that (1) visible DEM errors can occur in relatively flat terrains; and (2) DEM errors



caused by SAR-DEM misregistration depend on the amount of pixel misregistration, the local slope, and InSAR perpendicular baseline.

## 2 Methods

In this section, we first describe the conceptual model that relates the amount of active layer soil water to ground ice melting during summer thaw seasons (Section 2.1). We then explain our InSAR processing strategy for estimating average seasonal thaw subsidence from ALOS PALSAR data (Section 2.2), and discuss key error sources in InSAR measurements (Section 2.3). Finally, we review available field observations and strategies for validating the InSAR results (Section 2.4).

### 2.1 Estimating Soil Water Storage in the Saturated Active Layer from Thaw Subsidence Measurements

Our study site near Toolik Lake is in continuous permafrost of the upper Kuparuk River basin on the North Slope of Alaska (Figure 1). In 1987, the Toolik Field Station became part of the NSF Long Term Ecological Research program (LTER), which maintains long-term meteorological, ecological, and hydrological observations of the Arctic Foothills (Hobbie and Kling, 2014). The availability of the long-term databases of many basic parameters of the permafrost system makes the Toolik area an excellent site for studying how different soils control hydrological dynamics and may change as the climate warms and

permafrost thaws.

Based on in situ thaw measurements at Toolik, the active layer starts to thaw in early June, and the maximum seasonal thaw typically occurs in late August (Romanowicz and Kling, 2022). Thawing processes typically slow down around the time of maximum thaw, because (1) thermal diffusivity of ice is larger than that of liquid water; and (2) heat takes much longer to diffuse through a thicker active layer soil column. Due to the density difference between ice and liquid water, the land surface

subsides during the thaw season, with the opposite occurring when the active layer refreezes (Short et al., 2011; Painter et al., 2016; Sjoberg et al., 2016; Antonova et al., 2018; Strozzi et al., 2018). The maximum seasonal thaw subsidence ($d_{season}$) is proportional to the amount of water that experiences the ice-to-water phase change in a thaw season (denoted as $z_{water}$) following (Liu et al., 2012; Chen et al., 2020):

$$d_{season} = \frac{\rho_w - \rho_i}{\rho_i} z_{water} \approx 0.09 z_{water} \tag{1}$$

where $\rho_w$ and $\rho_i$ are the density of water and ice respectively. Here we exclude soil water stored above the water table (tension or unsaturated zone water) in the $z_{water}$ estimation. Because the porosity in the organic soil layers is high ($\sim 0.78$-$0.98$), water in the unsaturated zone can expand to fill the empty pore space during freezing without contributing to surface deformation. In this study, we assume the density of water is a constant value of 0.997 $g/cm^3$, and the density of ice is a constant value of 0.917 $g/cm^3$. Our calculation does not account for variations in subsurface water and ice density due to capillarity associated

with surface tension, cation hydration, surface hydration, and interlamellar cation hydration (Zhang and Lu, 2018).

Equation (1) shows that $d_{season}$ is proportional to $z_{water}$ rather than to the Active Layer Thickness (ALT). For example, minimal thaw subsidence signals would be observed over thick but dry active layers (Figure 2). This means that active layers



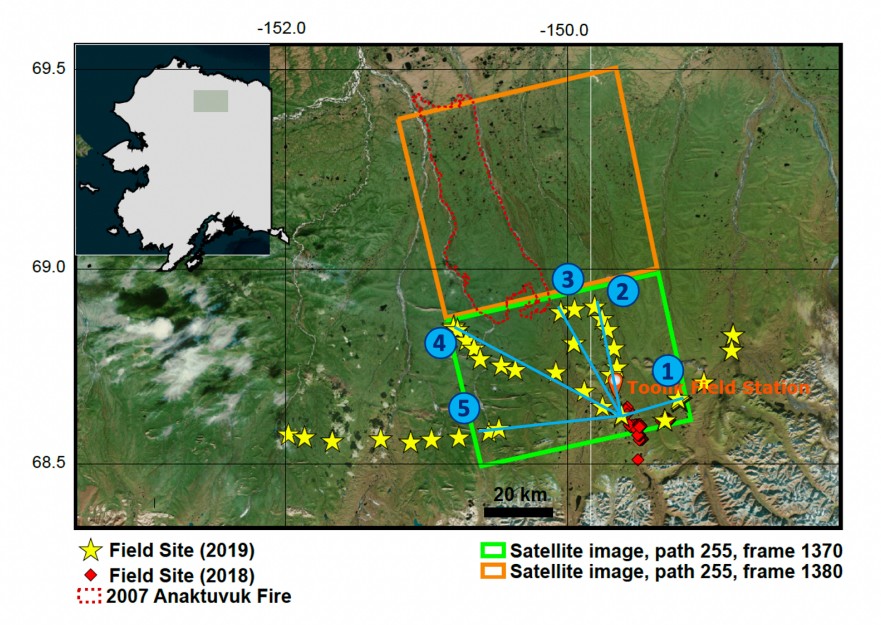

**Figure 1.** A map of the Toolik study site. The 2018 and 2019 sample sites are shown in red diamonds and yellow stars. The ALOS PALSAR coverage is outlined in green (path 255, frame 1370) and orange (path 255, frame 1380). The blue lines show five helicopter flight lines within the satellite data coverage, along which field measurements were collected. The 2007 Anaktuvuk River Fire scar is outlined with a red dashed line.

with higher ice-to-water content are expected to experience larger thaw subsidence, which may have no bearing on ALT. Furthermore, the active layer (liquid) water storage balance can be defined as:

$$\Delta S = A + (P - ET - Q) \tag{2}$$

where $\Delta S$ is the change in total soil water storage of the active layer. $P$, $ET$, and $Q$ stand for changes in soil water storage due to precipitation, evapotranspiration, and runoff, respectively. $A$ is the amount of soil water change associated with the active layer freeze and thaw process detectable by InSAR. When the active layer thaws during the summer, $A > 0$; when the temperature drops in autumn and the active layer refreezes, $A < 0$. In the case that InSAR-observed seasonal thaw subsidence signals are similar over multiple years, the amount of water that experienced the annual freeze-thaw cycle does not change much during this period (the net water drainage $P - ET - Q \approx 0$).

We emphasize that many geophysical processes can lead to surface deformation in permafrost terrain detectable by InSAR. For example, solifluction and other slope creep processes may produce long-term downward deformation trends in regions with large slope angles (Dini et al., 2019). Post-glacial rebound and tectonic motions typically vary at 100-km or larger spatial scales and can be considered as nearly spatially uniform over our study area (Liu et al., 2010; Stephenson et al., 2022). Given that InSAR measures relative deformation with respect to a local reference point, InSAR is only sensitive to spatially varied surface





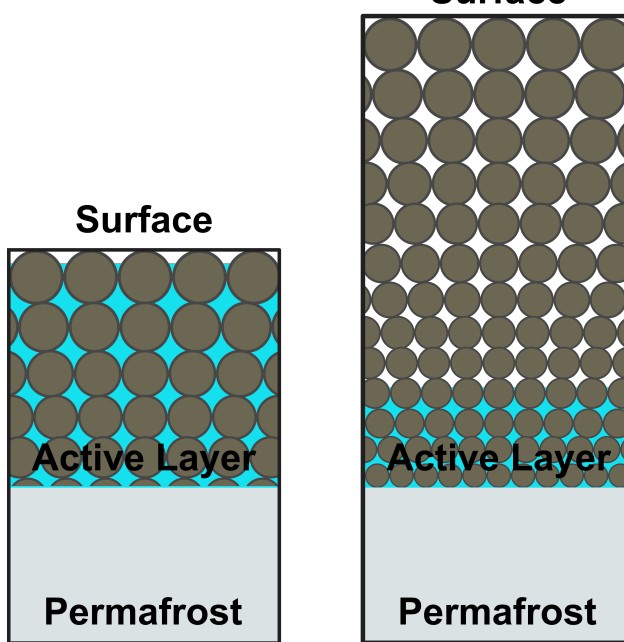

**Figure 2.** (Left) a nearly-saturated soil column with a shallower active layer. (Right) a relatively dry soil column with a thicker active layer. Because the left soil column stores more soil water in the active layer, we expect to observe more thaw subsidence over this soil column.

deformation over the study area. Hydrological loading and unloading can produce millimeter-level surface deformation signals (Liu et al., 2010), which is much smaller than centimeter-level freeze-thaw deformation. Furthermore, peat accumulation processes (Jones et al., 2017) may lead to a long-term deformation signal detectable by InSAR, and surface erosion can cause

changes in surface scattering properties that decorrelate radar phase measurements (Zebker and Villasenor, 1992). In Section 2.2, we discuss how to extract long-term and seasonal deformation signals from InSAR observations. The magnitude and characteristics of deformation signals, combined with in-situ observations (Section 2.4), can be used to determine the primary geophysical processes that contribute to the observed deformation patterns.

### 2.2 InSAR Processing Strategy

Interferometric SAR (InSAR) computes the phase difference between two Synthetic Aperture Radar (SAR) images. The resulting interferogram can be used to infer a map of surface deformation between two SAR acquisition times along the radar Line-Of-Sight (LOS) direction (Hanssen, 2001). More specifically, a phase cycle of $2\pi$ (in radians) equals to $\lambda/2$ of LOS deformation, where $\lambda$ is the radar wavelength. For L-band ALOS PALSAR data, $\lambda$ equals 24 cm, and thus a phase cycle of $2\pi$ represents 12 cm of LOS deformation occurred between two radar acquisition times.

In a recent proof-of-concept study (Chen et al., 2020), we processed 12 L-band ALOS PALSAR scenes (Table A1) acquired during summer seasons (June to October) between 2006 and 2010 from path 255 frame 1370 over our study region (Figure





1). Note that we excluded all winter scenes acquired between November and April because the observed phases in winter-winter interferograms are likely related to variations in snow accumulation and snow redistribution, which is not the focus of this study. We first solved for the long-term LOS deformation trend at a pixel of interest based on a stacking approach (Sandwell and Price, 1998; Lyons and Sandwell, 2003; Rouyet et al., 2019). That is, averaging all interferograms that contain minimal seasonal deformation signals (e.g., a September-to-September pair) and relatively large long-term signals (e.g., span multiple freeze-thaw cycles). An important finding of this pilot study was that no detectable long-term deformation trend above the InSAR measurement noise level was observed outside the 2007 Anaktuvuk River fire scar (Figure 1) during the study period of 2006 to 2010. This allowed us to substantially simplify our InSAR processing strategy for reconstructing seasonal freeze-thaw deformation patterns over undisturbed permafrost terrain. We estimated the LOS deformation signatures due to the seasonal active layer freeze-thaw processes between (i) early June and late July, (ii) late July and early September, and (iii) early September and late October by averaging all interferograms that span these periods regardless of how many years those interferograms span. The averaged LOS deformation between early June and late July was used as an approximation of the maximum seasonal LOS deformation because no ALOS acquisitions were made over the study area around the time of the maximum thaw (late August at Toolik area). This approximation is reasonable given that a few centimeters of late summer thaw (August) of the relatively dry low-porosity mineral layer does not cause surface thaw subsidence detectable by InSAR (O'Connor et al., 2020; Chen et al., 2020). For example, 10 cm of thaw of the saturated organic layer ($\sim 90\%$ porosity) would lead to $\sim 0.8$ cm of thaw subsidence. In comparison, 10 cm of thaw of the saturated mineral soil ($\sim 20\%$ porosity) would lead to $< 0.2$ cm of thaw subsidence.

We note that InSAR measures the change in distance between the antenna and the ground object, known as the LOS direction. Assuming the horizontal motion of the land surface is negligible, we converted InSAR seasonal deformation estimates along the LOS direction ($d_{LOS}$) to seasonal vertical thaw subsidence estimates $d_{season}$ as:

$$d_{season} = \frac{d_{LOS}}{e_3} \tag{3}$$

where $e_3$ is the vertical component of the radar LOS direction unit vector $e = [e_1, e_2, e_3]$. The LOS unit vector $e$ can be computed based on the known satellite position and ground pixel location in the Earth-centered, Earth-fixed (ECEF) coordinate system, and then converted to the local east-north-up (ENU) system (Misra and Enge, 2011). For the ALOS ascending imaging geometry over the Toolik area, $e = [0.61, 0.13, -0.78]$ at the mid-swath, and the variation of $e_3$ across the entire swath is minimal (less than 3%). This means that $\sim 5$ cm thaw subsidence can cause 4 cm positive LOS deformation for the Toolik ALOS PALSAR case.

To confirm that our InSAR processing strategy is suitable for studying the active layer freeze-thaw process over vast areas, here we analyzed an additional 11 L-band ALOS PALSAR scenes (Table A1) acquired during summer seasons (June to October) between 2006 and 2010 from path 255 frame 1380 (Figure 1). We merged interferograms from the same path but two different frames by calibrating the phase differences within the overlapping regions of the two frames. A sample merged interferogram is shown in Figure A1, and the same reference point (68.83° N, 150.23° W) as our previous study (Chen et al., 2020) was used to calibrate all interferograms. We chose this reference point because it is in a dry highland area of relatively flat





terrain, and the expected seasonal deformation is minimal. Only $\sim 4\%$ of interferograms contain visible phase decorrelation artifacts outside the fire scar, and the overall phase coherence (Figure B1(a)) of the remaining interferograms is comparable to the sample interferogram (Figure A1). We masked out low amplitude and low phase coherence pixels that include water bodies and the area burned by the 2007 Anaktuvuk River fire (Figure B1(b)). A comparable pixel mask can also be generated using

the North Slope Science Initiative (NSSI) land cover GIS Data. Similar to our previous study, we found that the long-term subsidence trend is negligible outside the fire scar (Figure C1(d)). This allows us to follow the same processing strategy as our previous study to extract seasonal deformation between (i) early June and late July, (ii) late July and early September, and (iii) early September and late October by averaging all interferograms that span these periods regardless of how many years those interferograms span (Figure C1(a)-(c)). We note that averaging interferograms that contain the common signal of interest

(stacking) reduces the impact of random noise by $\sim \sqrt{N}$, where $N$ is the number of independent SAR acquisitions (Sandwell and Price, 1998; Chen et al., 2020). A thaw subsidence pattern similar to the final stacking solution was identified from all individual interferograms that span a common season (e.g., early June to late July).

Based on Equation (1) and Equation (3), we further established a linear relationship between InSAR LOS deformation observations and the amount of water in the active layer that experiences the ice-to-water phase change in a thaw season

($z_{water}$):

$$z_{water} = \frac{\rho_i}{(\rho_w - \rho_i)e_3} d_{LOS} \tag{4}$$

where $\rho_w$ and $\rho_i$ are the density of water and ice, respectively. This equation shows that InSAR-observed seasonal thaw subsidence is proportional to the active layer water storage $z_{water}$. For the ALOS Toolik case ($e_3 = -0.78$), 5 cm InSAR LOS deformation measurements ($d_{LOS}$) can be related to 70 cm of saturated active layer soil water column ($z_{water}$), 1 cm errors

in InSAR LOS deformation measurements can lead to 14 cm error in $z_{water}$ estimates. We note that Equation (4) employs the assumption that the horizontal motion of the land surface is negligible. Our study site is a transitional region located between the Coastal Plain and the steep mountains of the Brooks Range, which consists of gently rolling hills and broad exposed ridges that extend along the northern flank of the Brooks Range. Given that the long-term subsidence trend is negligible outside the fire scar and seasonal deformation signatures follow the expected seasonal freeze-thaw patterns (Figure C1), InSAR observations

at our study site are primarily related to the volume change associated with water-to-ice phase change rather than slope creep processes. For a 5% slope angle, 1 cm of freeze-thaw deformation perpendicular to the land surface leads to 0.87 mm horizontal deformation and 9.96 mm vertical deformation. For a 10% slope angle, 1 cm of thaw deformation perpendicular to the land surface leads to 1.74 mm horizontal deformation and 9.85 mm vertical deformation. Because the slope angle at most radar pixels is less than 10%, we conclude that the assumption of negligible horizontal motion is reasonable.

## 2.3   Error Sources in InSAR-based $z_{water}$ Estimates

To quantify errors in InSAR-based $z_{water}$ estimates, here we evaluate major error sources in InSAR LOS deformation solutions ($d_{LOS}$), which can be written as (Zebker and Villasenor, 1992; Zebker et al., 1994, 1997):

$$d_{\mathrm{LOS}} = \frac{\lambda}{4\pi}\varphi + \Delta d_{dem} + \Delta d_{decor} + \Delta d_{unwrp} + \Delta d_{orb} + \Delta d_{atm} + \Delta d_{iono} + \Delta d_n \tag{5}$$





where $\lambda$ is the radar wavelength (24 cm for L-band ALOS data), and $\varphi$ is the average phase of all interferograms that contain the common seasonal deformation signal of interest. The remaining noise terms on the right-hand side are errors due to topography-related artifacts ($\Delta d_{dem}$), phase decorrelation ($\Delta d_{decor}$) and phase unwrapping errors ($\Delta d_{unwrp}$), orbital errors ($\Delta d_{orb}$), atmospheric ($\Delta d_{atm}$) and ionospheric ($\Delta d_{iono}$) artifacts, and other smaller error terms associated with thermal and soil moisture effects ($\Delta d_n$).

In the Toolik ALOS InSAR data analysis, we excluded $\sim 4\%$ of interferograms containing visible phase decorrelation and phase unwrapping errors. Long-wavelength phase signatures, varying at spatial scales of tens to hundreds of kilometers and potentially caused by orbital errors and tropospheric or ionospheric noises, were removed as a planar phase ramp from each interferogram (Staniewicz et al., 2020; Wang and Chen, 2022; Zebker et al., 2023). The deramp process does not remove localized freeze-thaw deformation patterns that vary from hilltop ridges to the lowland valleys and riparian zones on the spatial scale of $\sim$ 100s of meters. Interferograms formed by the SAR scenes acquired on 8 September 2008 (for frame 1380) and SAR scenes acquired on 22 July 2007, 24 July 2008, and 14 September 2010 (for both frame 1370 and 1380) were also excluded because of severely distorted ionospheric artifacts (Gray et al., 2000; Wegmuller et al., 2006; Chen and Zebker, 2012; Fattahi et al., 2017). Because of a cool and dry tundra climate and relatively small elevation variation ($\sim$ 200-300 meters), the stratified tropospheric noise component (Doin et al., 2009) is minimal over the study site. Given that long-wavelength tropospheric and ionospheric noise was removed during the planar ramp removal process, the residual atmospheric noise term (e.g., due to localized temperature or water vapor variations) is mostly random at time scales longer than one day, but is correlated in space and typically increases with distance from the InSAR reference point (Emardson et al., 2003; Staniewicz et al., 2020). Assuming a 2 cm tropospheric error in each ALOS PALSAR interferogram (Zebker et al., 1997; Emardson et al., 2003), the turbulent random noise level can be reduced to less than 1 cm after stacking four interferograms formed from four SAR acquisitions. In the remainder of this section, we focus on the dominant error term associated with topography-related artifacts for the ALOS Toolik case.

At a pixel of interest, an error in the Digital Elevation Model (DEM; $\delta$) with respect to the reference pixel can lead to an error ($\Delta d_{dem}$) in the LOS deformation estimates as (Berardino et al., 2002; Werner et al., 2003; Fattahi and Amelung, 2013):

$$\Delta d_{dem} = \frac{B_{perp}}{r sin\theta_l}\delta \tag{6}$$

where $B_{perp}$ is the perpendicular component of the InSAR spatial baseline, which can be calculated from known radar imaging geometry. For ALOS interferograms, $B_{perp}$ typically ranges from several hundred to several thousand meters. $r$ is the distance between the radar antenna and the ground pixel, and $\theta_l$ is the radar look angle. Because the look angle of ALOS PALSAR does not vary much over the $\sim$ 60 km radar swath, both $r \sim 850$ km and $\theta_l \sim 34$ degrees can be approximated as constant values for all ALOS interferograms collected from the same path and frame.

In this study, we removed the topographic phase during interferogram formation using the Arctic DEM v3.0 (10-meter resolution and resampled to a 30-meter grid) data (Porter et al., 2018), which are widely used in the Arctic community because of its pan-arctic coverage and high quality (Tozer et al., 2019). Interferograms with comparable quality can be also generated using the Kuparuk River watershed DEM (Chen et al., 2020). While the Kuparuk River watershed DEM has been thoroughly





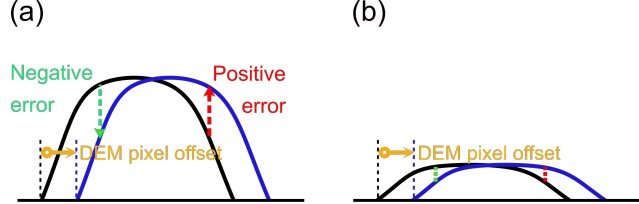

**Figure 3.** An illustration of SAR and DEM misregistration in hilly terrains. The DEM profile is shown in black. When the SAR image (blue) is misregistered by 1 pixel to the east, there is a positive DEM error on the east-facing slope and a negative DEM error on the west-facing slope. Similarly, pixel misregistration to the north or south can lead to DEM errors on the north- and south-facing slopes across the hill ridge. The same amount of pixel misregistration leads to larger errors in areas with steep terrain (panel a) than in relatively flat terrain (panel b).

validated and highly accurate (Nolan, 2003b), it does not have complete spatial coverage over the entire study area. It is common to assume that $\Delta d_{dem}$ is linearly proportional to $B_{perp}$. This assumption is valid when $\delta$ in Equation (6) is introduced by

errors in the DEM dataset itself (thus $\delta$ is the same for all interferograms). Because thaw subsidence patterns over undisturbed permafrost terrain are expected to be spatially coherent, phase discontinuity in interferograms was visually inspected. If the magnitude of these artifacts is linearly proportional to InSAR perpendicular baseline $B_{perp}$, they are likely associated with the errors in the Arctic-DEM dataset, given that thaw subsidence signals do not depend on $B_{perp}$. Furthermore, we discovered that $\delta$ can also be introduced by misregistration of the DEM and a SAR image. For example, as shown in Figure 3(a), a radar image

(blue) and a topography map (black) are misregistered by 1 pixel to the east, which leads to a positive $\delta$ on the east-facing slope and a negative $\delta$ on the west-facing slope. Similarly, pixel misregistration to the north or south can lead to $\delta$ on the north- and south-facing slopes across the hill ridge. Because the same amount of pixel misregistration leads to larger $\delta$ in areas with larger slopes (Figure 3(b)), these artifacts are most prominent in interferograms formed using misregistered SAR scenes over steep terrains. To better understand these pixel-mismatching artifacts, we approximated the DEM error $\delta$ due to pixel misregistration

as the difference between the Arctic-DEM and the shifted Arctic-DEM in east/west and north/south directions. For example, the DEM error $\delta_{i,j}$ due to 1 pixel misregistration to the east at pixel $(i, j)$ can be written as:

$$\delta_{i,j} = h_{i,j} - h_{i+1,j} \tag{7}$$

where $h_{ij}$ is the Arctic-DEM at pixel $(i, j)$. Similarly, we can approximate $\delta_{i,j}$ due to 1 pixel misregistration to the south at pixel $(i, j)$ as $h_{i,j} - h_{i,j+1}$. We then calculated the expected LOS errors $\Delta d_{dem}$ due to $\delta$ based on Equation (6) for a given

imaging geometry and perpendicular baseline. Results from these numerical experiments were then compared to actual InSAR LOS observations across hill ridges.

## 2.4 Field Observation for validating InSAR-estimated $z_{water}$

Our InSAR thaw subsidence estimates were validated using a relatively large number of field observations collected within ∼ 100 km of Toolik Field Station (Figure 1) in 2018 (August 15 - August 24) and 2019 (July 26 - August 3). Particularly, the

amount of water in the active layer can be quantified by determining saturated active layer thickness and porosity. Tundra soil





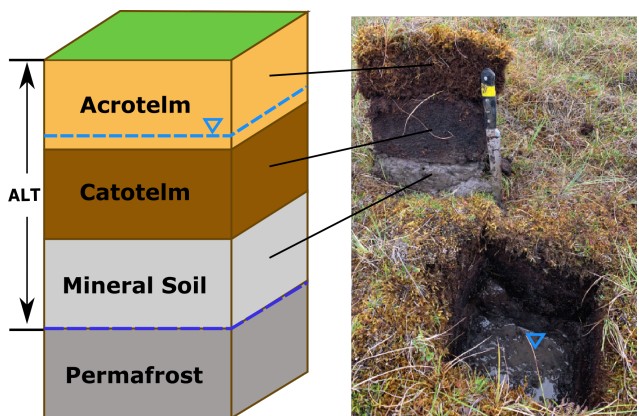

**Figure 4.** (Left) A cartoon showing the three soil layers. The lighter blue dashed line denotes the groundwater level, and the darker blue line shows the location of the permafrost table note that the depth of permafrost may be in any soil layer. (Right) A photo of a soil pit with three soil layers from top to bottom: the acrotelm (peat that contains living plants), the catotelm (peat that contains dead plant materials), and the mineral soil.

in the Toolik area consists of three layers from top to bottom: the acrotelm (peat that contains living plants), the catotelm (peat that contains dead plant materials), and the mineral soil (Figure 4). The thickness of these three soil layers and the depth to the water table were measured at each sampling site. The porosity ($\phi$) of each soil core sample was also measured to characterize the water-holding capacity of active layer soils. $z_{water}$ can then be calculated as:

$$z_{water} = \sum_{i=1,2,3} z_{si}\phi_i \tag{8}$$

where $z_{si}$ and $\phi_i$ are the saturated thickness and the porosity of the $i^{th}$ soil layer. Here, we assume the soil column below the water table is fully saturated. We also note that the mineral soil layer has much lower porosity (thus much less water-holding capability) than organic soil layers. For example, a fully saturated, 10-cm-thick acrotelm layer with a porosity of 0.90 contributes to 9 cm of $z_{water}$, while a fully saturated, 10-cm-thick mineral soil layer with a porosity of 0.20 only contains 2 cm of $z_{water}$.

To jointly analyze remote sensing and in situ observations, an exact point-to-point comparison is challenging, if not impossible, because they were collected at very different spatial and temporal scales. A pixel in an InSAR-derived deformation map is on the order of 10s to 100s of meters, while field measurements were collected at sites with size $\sim 1000\ cm^2$ (30-by-30 cm plots). To overcome this challenge, we designed a statistical comparison approach. This was done by fitting probability density functions (PDFs) to the empirical distributions (histograms) of the in situ soil property measurements, including the thickness and porosity of the acrotelm, the catotelm, and the mineral soil as well as the depth to water table (O'Connor et al., 2020; Chen et al., 2020), and using these distributions to calculate the range of possible thaw subsidence. There are also sources of error in the property measurements, which are (1) errors from reading the measured value, which is typically small (e.g., <0.5 cm for thaw depth measurements from probing), and (2) in situ measurements varying due to the sub-meter-scale heterogeneity of





arctic soils (e.g., waviness of the ice-table). Nonetheless, we found that the fitted PDFs stayed mostly the same as the sample
size increased after a second season of sampling in 2019, indicating that the sample size in this study was sufficiently large to
capture the statistics of the soil properties at the regional scale. We drew random samples from the PDFs of soil properties, and
calculated the statistical distribution of $z_{water}$ following Equation (8).

Finally, we validated InSAR-observed $z_{water}$ using field-based predictions of $z_{water}$ at different vegetation types. For the

purpose of studying active layer soil properties, we grouped various subclassifications of vegetation types over the study area
(Walker and Walker, 1996; Stow et al., 2004; Walker et al., 2017) into four primary land cover types: "sedge", "tussock",
"woody shrub", and "sparse vegetation" (O'Connor et al., 2020). The sedge land cover typically occurs in wet to saturated
sites (e.g., riparian zones) and may occasionally mix with shrub mounds on slightly elevated ground. The tussock land cover is
distributed broadly from ridges to riparian zones. The term "woody shrub land cover" refers to areas dominated by woody-stem

plants, which include both woody shrubs along the water tracks and heath vegetation on ridges. Because the soil in water tracks
typically consists of well-drained acrotelm with underlying gravel and boulders, we did not collect soil samples in the water
tracks. As a result, this study focuses on soil measurements collected over three land cover types: sedge, tussock, and woody
shrub on ridge-tops (referred to as "heath"). Photographs of the land cover types are shown in Figure D1. We also identified
the land cover type of each InSAR pixel using the North Slope Science Initiative (NSSI) Land Cover Map (Payne et al., 2016),

which does not distinguish between woody shrubs within water tracks and heath on hill ridges.

## 3 Results and Discussion

### 3.1 InSAR-estimated soil water depth in the saturated active layer

InSAR-observed average seasonal thaw subsidence estimates (2006-2010) between early June and late July from two independently processed ALOS PALSAR frames are consistent with no visual discontinuity or artifacts (Figure 5(a)). This confirms

that our InSAR analysis is robust for reconstructing thaw subsidence over permafrost terrain. Ninety-five percent of the observed thaw subsidence ranged from 0 cm to 5.4 cm, which correlates with the topography as well as the watershed and river
network morphology (Figure 5(b)). The drier ridge-top areas usually show less than 2 cm thaw subsidence, while the wetter
valleys and riparian zones show up to 6 cm subsidence. Thaw subsidence of $\sim 4$ cm is observed near the transition zone as the
steeper hilly terrain (south) transitions to flatter plains (north). Based on Equation (1), 1 cm thaw subsidence ($\sim 0.78$ cm LOS

deformation) is caused by an $\sim 11$ cm soil water column that experiences the ice-to-water phase change in the saturated active
layer (denoted as $z_{water}$). Ninety-five percent of $z_{water}$ estimates range from 0 to 62 cm in the Toolik area, with up to 75 cm
$z_{water}$ observed in the wettest riparian zone after removing less than 3% of outliers (Figure 5(c)). Here, pixels are marked as
outliers if they are larger than the upper adjacent value, which is, by definition, the largest observation that is less than or equal
to the threshold located at the 1.5 Inter-Quartile Range (IQR) above the upper quartile (Q3). The spatial variation of $z_{water}$

is consistent with groundwater flows and accumulation from the higher ridges to the flatter riparian zones and valleys (e.g.,
Figure 6). Large $z_{water}$ values are often observed in wet local low regions.



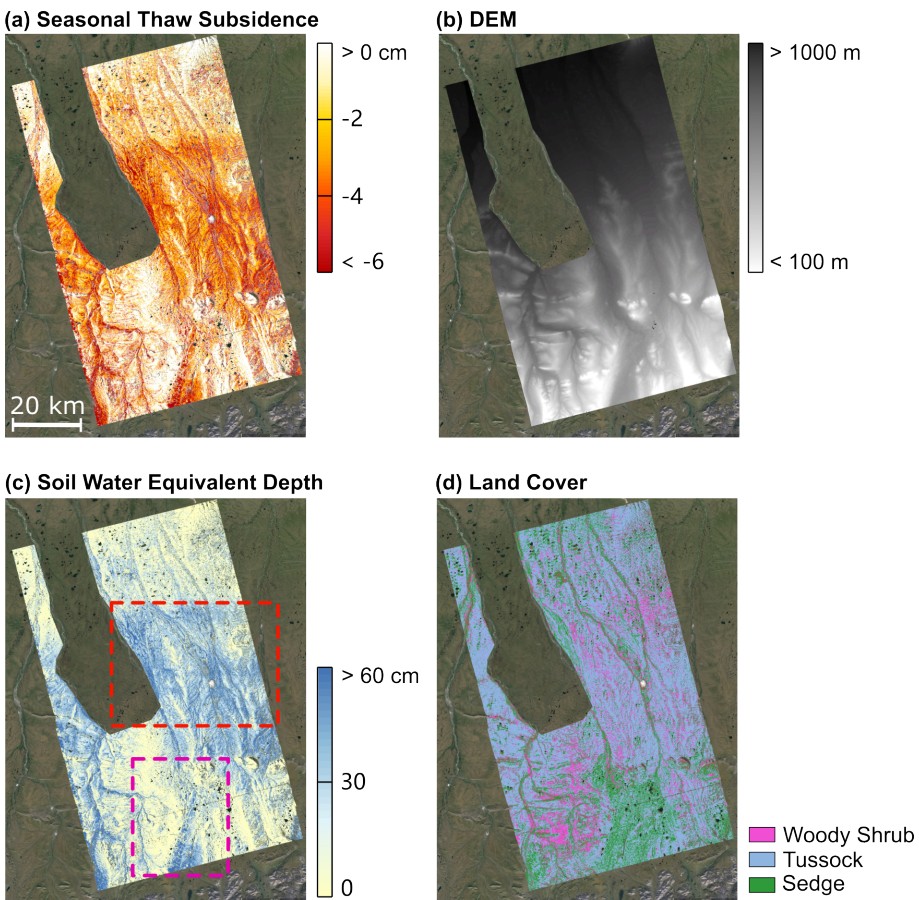

**Figure 5.** (a) Seasonal thaw subsidence (in the vertical direction) over the area of interest. A darker red color means larger thaw subsidence between early June and late July during the 2006-2010 study period. Water bodies and the area burned by the 2007 Anaktuvuk River fire have been masked out. (b) Digital Elevation Model of the same region. A darker color indicates a lower elevation. (c) InSAR-estimated saturated active layer soil water storage ($z_{water}$) map. A darker blue color indicates a larger amount of $z_{water}$. (d) Land vegetation cover map of the same region.



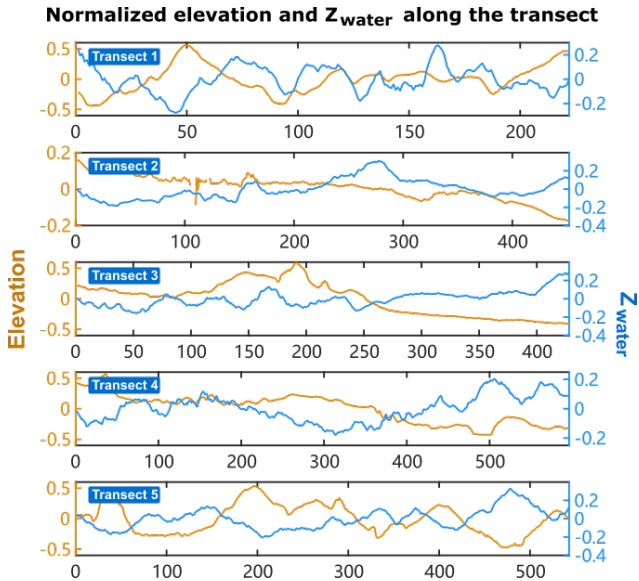

**Figure 6.** Normalized surface elevation (yellow) and InSAR-estimated saturated active layer soil water storage (blue) along five flight lines shown in Figure 1. The original elevation and $z_{water}$ data were adjusted to a notionally common scale by subtracting the mean from the original data and dividing the data by its range. The normalized $z_{water}$ curve was then smoothed using a box car filter with a window size equal to 4% vector length to match the spatial resolution of the elevation data.

Because the amount of soil water influences the type of vegetation that can grow, the spatial pattern of InSAR-observed $z_{water}$ correlates with land cover types (Figure 5(d)). We found that land cover type indicates the characteristics of soil stratigraphy (Figure E1), and each soil layer possesses different characteristics (e.g., porosity and thickness) that influence the water-holding capability of the active layer. For example, water-loving sedges tend to grow on wet soils with a thick porous catotelm layer and a shallow water table, while heath vegetation is often found on dry hill ridges with a thin catotelm layer and a deep water table. To further illustrate the spatial correlation between the amount of soil water and land cover types, Figure 7 (a)-(c) shows a zoomed-in area near the Toolik Field Station from frame 1370 (with the location outlined by the purple dashed line in Figure 5(c)), where all three major land cover types are present. We found that (1) sedges are often distributed over regions with large $z_{water}$ values, where open water bodies are visible in the optical image; (2) woody shrubs are typically distributed over well-drained high ground. On average, soils covered by sedges store 23% more water than soils covered by tussocks and 58% more water than soils covered by woody shrubs (Table 1). Figure 7 (d)-(f) shows another zoomed-in area from frame 1380 (with the location outlined in a red dashed line in Figure 5(c)), where the terrain transitions from rolling hills to coastal plains. This region is wetter than the Toolik Lake area (Figure 7 (g) and Table 1). Here, tussock is the dominant land-cover type, and water-loving shrubs and sedges are distributed along the water tracks (visible in the optical image).

To validate InSAR $z_{water}$ estimates, the expected distribution of $z_{water}$ was also calculated from field measurements collected near Toolik (Figure 1) following Equation (8). We found that $z_{water}$ estimated from field and satellite observations is





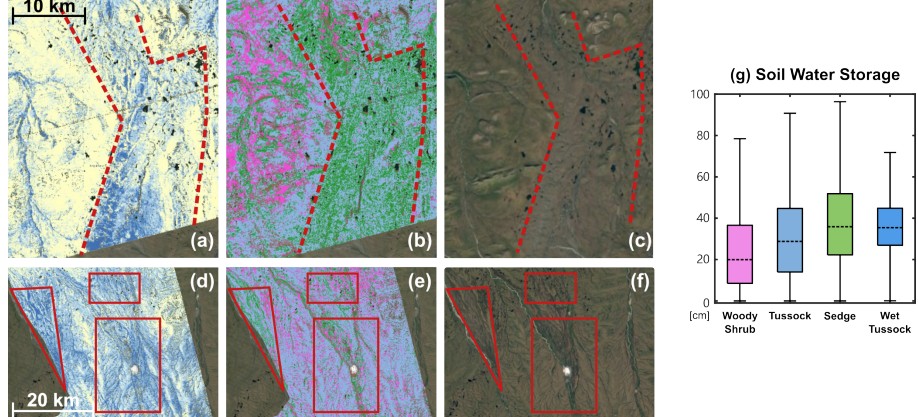

**Figure 7.** (a-c) InSAR-estimated saturated active layer soil water storage ($z_{water}$) map, land cover map, and optical image (from Google Earth Landsat imagery acquired in April 2013) for the Toolik area (outlined in purple dashed line in Figure 5(c)). (d-f) InSAR-estimated saturated active layer soil water storage ($z_{water}$) map, land cover map, and Landsat optical image (provided by Google Earth) for the northern study area (outlined in red dashed line in Figure 5(c)). Areas outlined in red show larger $z_{water}$ values. The color bar and legend are the same as Figure 5(c) and (d). (g) Boxplots of InSAR-estimated $z_{water}$ for woody shrub, tussock, and sedge in the Toolik area (outlined in purple dashed line in Figure 5(c)), and wet tussock in the northern study area (outlined in red dashed line in Figure 5(c)). The wet tussock in the north generally stores more soil water than those growing near the Toolik area. The boxplots display lower adjacent, lower quartile, median, upper quartile, and upper adjacent values.

**Table 1.** Relationship between soil water equivalent depth and land vegetation cover

| Unit: cm | Woody Shrub | Tussock | Sedge | Wet Tussock |
|---|---|---|---|---|
| Mean | 24.70 | 31.82 | 38.98 | 36.61 |
| Std. dev. | 20.26 | 23.00 | 23.37 | 14.95 |
| Q1 - 25th % | 8.47 | 13.96 | 22.23 | 26.89 |
| Median - 50th % | 19.95 | 28.78 | 35.89 | 35.43 |
| Q3 - 75th % | 36.63 | 44.74 | 51.88 | 44.85 |





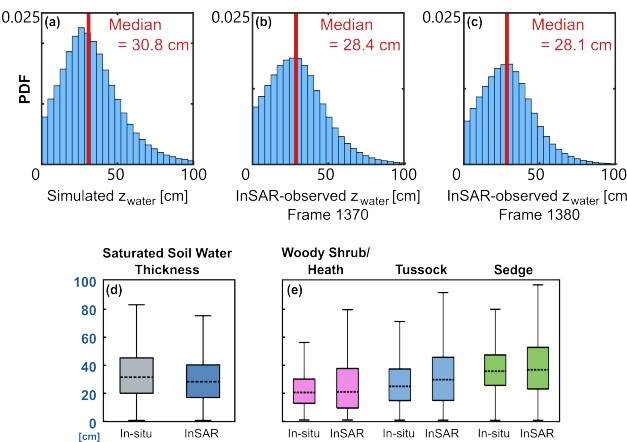

**Figure 8.** Histograms of saturated active layer soil water storage $z_{water}$ calculated from (a) in-situ data, and (b)-(c) ALOS PALSAR path 255 frame 1370-1380 thaw subsidence estimates. InSAR pixels with a long-term trend greater than 6 mm/year were excluded in the statistics, because this study is focused on undisturbed permafrost terrain with negligible subsidence trend. (d) Boxplots comparing $z_{water}$ calculated from in situ data and InSAR thaw subsidence estimates. (e) Boxplots comparing $z_{water}$ calculated from in situ data and InSAR thaw subsidence for woody shrub (heath), tussock and sedge land covers. Here $z_{water}$ derived from InSAR subsidence observations is for the Toolik area in Figure 7 (a). The boxplots display lower adjacent, lower quartile, median, upper quartile, and upper adjacent values.

statistically consistent (Figure 8(a)-(d)). The median $z_{water}$ values derived from field data, ALOS PALAR path 255 frame 1370 data, and ALOS PALSAR path 255 frame 1380 data are 30.8 cm, 28.4 cm, and 28.1 cm, respectively. The standard deviation

values of $z_{water}$ derived from field data, ALOS PALAR path 255 frame 1370 data, and ALOS PALSAR path 255 frame 1380 data are 21.2 cm, 18.9 cm, and 17.3 cm, respectively. Both InSAR and field observations are also consistent over three major land cover types (Figure 8 (e)). Based on in situ data, $z_{water}$ has a median of 19.5 cm, 24.1 cm, and 34.9 cm for heath, tussock, and sedge land covers. This is consistent with InSAR observations over three land cover types: 20.0 cm for woody shrubs, 28.8 cm for tussocks, and 35.9 cm for sedges (Table 1). InSAR observations for each land-cover type generally show a larger

variation of $z_{water}$ compared to field observations. This is likely because InSAR pixels were classified using the NSSI land cover map, which is less accurate than field-based land-cover classification at each sampling site. We also note that the land cover map used for classifying InSAR pixels does not distinguish woody shrubs located near the water tracks and those on dry ridge tops, while during field data collection, we only sampled dry heath land covers on the ridges. This is another reason that InSAR woody shrub observations show a larger variation compared to the other two land cover types.

Due to the remote nature of the study area, the number of available field observations is limited. We acknowledge that most field sites are located within the coverage of ALOS PALSAR path 255 frame 1370. Nevertheless, both frames exhibit similar land cover type combinations, suggesting similar climatic and geological settings (Figure 5 (d)). While some of our field sites are located outside the radar footprint (Figure 1), field observations at these sites follow similar statistical distributions as those sites located within the radar footprint. Furthermore, InSAR-observed average seasonal thaw subsidence estimates (2006-2010)

from two independently processed ALOS PALSAR frames are consistent with no visual discontinuity or artifacts at the frame





boundary (Figure 5 (a)). This indicates the InSAR processing strategy produced consistent thaw subsidence estimates. We observed surface subsidence due to the thawing of the active layer from early June to late July and a net surface uplift between late July and late October resulting from the refreezing of the soil (Figure C1(a)-(c)). Minimal long-term subsidence trends were observed outside the fire scar (Figure C1(d)). These observations confirm that the observed InSAR seasonal deformation signals at our study site are primarily related to the volume change associated with water-to-ice phase change. We translated
InSAR measurements to the soil water equivalent depth following Equation (4), which does not require additional information on soil properties. We used in-situ soil measurements as an independent validation for InSAR results in regions wherever it is possible, and our goal is to develop a remote sensing technique that can fill the observational gaps in remote Arctic areas with no in situ observations.

We also acknowledge that field observations were collected in 2018 and 2019, while ALOS PALSAR InSAR data were used to estimate the average seasonal thaw subsidence between 2006 and 2010. In-situ thaw depth measurements show that the August 11 thaw depth ($\sim$ 40 cm) at the Toolik long-term monitoring site has increased very slightly since 1990. At the Imnavait site, the August 11 thaw depth increase between 2006 and 2010 is $\sim$ 5 cm. At both sites, we did not observe any long-term subsidence trend above the InSAR noise level (Chen et al., 2020). This is because a 5 cm thaw of the low porosity
(thus less water-holding capacity) mineral soils was unlikely to cause any soil water content increase that is detectable by InSAR. Therefore, our study focuses on the comparison between InSAR average seasonal thaw subsidence estimates (2006-2010) and recent field observations over undisturbed permafrost terrain (relatively stable with long-term changes undetectable by InSAR). Due to the limited ALOS PALSAR data availability, the investigation of inter-annual variability of InSAR thaw subsidence patterns is outside the scope of this work.

## 3.2   The signature of Arctic-DEM errors

Because DEM error is the dominant error source in the ALOS Toolik InSAR data, here we discuss errors in thaw subsidence estimates associated with errors in the DEM data. When there is an error in the DEM data, a similar signature may be observed in the InSAR surface deformation observations. For example, Figure 9(a) shows seasonal thaw subsidence between early June and late July outside the 2007 Anaktuvuk River fire zone inferred from an L-band ALOS interferogram that spans 3 June 2006
and 30 July 2010. We zoomed into the region outlined in blue (Figure 9(c)), where a discontinuity in thaw subsidence estimates across the blue dashed line is visible and affects a relatively flat area. This artifact is likely associated with a discontinuity observed at the same location in the Arctic v3.0 Pan-Arctic DEM data (Figure 9(e)), which were used to remove topographic phases in interferograms. As a comparison, Figure 9(b) shows the seasonal thaw subsidence map between early June and late July over the same region inferred from an L-band ALOS interferogram that spans 8 June 2008 and 30 July 2010. While
both interferograms suggest similar seasonal thaw subsidence patterns, the error in the DEM data does not lead to any visible discontinuity in the thaw subsidence derived from the second interferogram (Figure 9(d)). This is because the perpendicular baseline is 5070 m for the interferogram shown in Figure 9(a) and 1558 m for the interferogram shown in Figure 9(b). As shown in Fattahi and Amelung (2013), DEM errors in the LOS measurement are linearly proportional to the perpendicular baseline for a fixed error in the Arctic-DEM (Equation (6)).





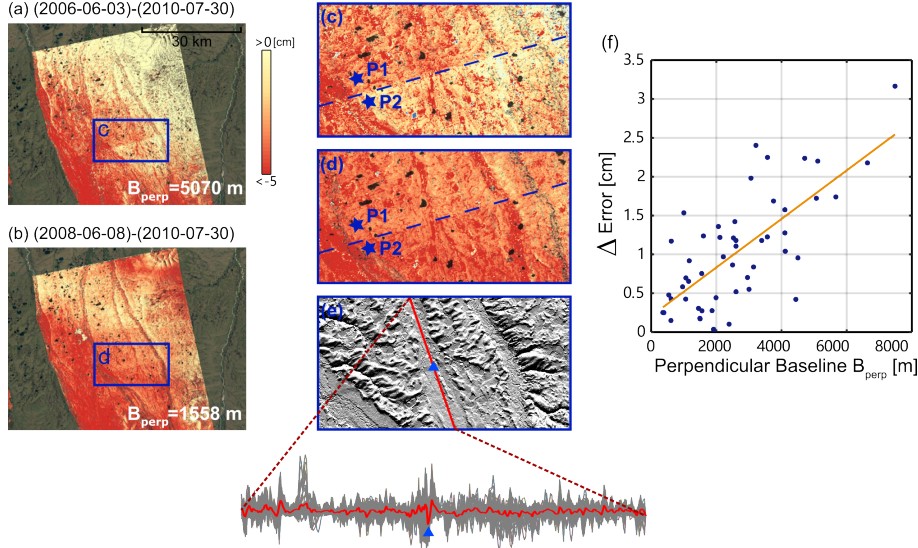

**Figure 9.** Seasonal thaw subsidence between early June and late July (cm; in orange-red color) outside the 2007 Anaktuvuk River fire zone as inferred from an L-band ALOS interferogram that spans (a) 3 June 2006 and 30 July 2010, and (b) 8 June 2008 and 30 July 2010. The perpendicular baseline is 5070 m for interferogram (a) and 1558 m for interferogram (b). (c) A zoomed-in region of interferogram (a) as outlined in blue. Here Arctic-DEM errors lead to a visible discontinuity in thaw subsidence estimates across the blue dashed line. (d) A zoomed-in region of interferogram (b) as outlined in blue. With a smaller perpendicular baseline, no substantial discontinuity exists across the blue dashed line. (e) A shaded relief map derived from the Arctic v3.0 Pan-Arctic Digital Elevation Model. The averaged DEM gradient of 50 transects along the red solid line shows the location of the Arctic-DEM error marked by the blue triangle symbol. This sharp discontinuity is co-located with the dashed line marked in panel (c). (f) Thaw subsidence difference (in cm) between P1 and P2 of all available interferogram pairs vs. the perpendicular baseline (in m). A linear relationship between the perpendicular baseline and the deformation error (orange line) can be observed.

To better illustrate that our observations are consistent with existing InSAR DEM error studies, we analyzed all 51 interferograms from path 255 frame 1380 and identified a linear relationship between the InSAR perpendicular baseline and the thaw subsidence errors at P1-P2 across the discontinuity line (marked in Figure 9 (c) and (d)). The observed linear slope (Figure 9 (f)) suggests a 1.16-meter error in the Arctic-DEM, which is consistent with the $\sim$ 1-2 m DEM discontinuity observed in the actual Arctic-DEM data (Figure 9 (e)). Existing InSAR studies tend to assume non-negligible DEM artifacts are typically
observed in areas with steep terrain (Li et al., 2014; Staniewicz et al., 2020; Zhou et al., 2020). However, our results show that more than 1.5 cm LOS errors associated with inaccurate DEM can be observed over relatively flat areas in ALOS PALSAR interferograms with $B_{perp}$ values > 4000 m. This error is on the same order of magnitude as the observed centimeter-level thaw subsidence signals; thus, it is not negligible. These artifacts caused by errors in the Arctic-DEM can be mitigated by (1) excluding interferograms with larger $B_{perp}$ or (2) estimating and removing a phase component that is proportional to $B_{perp}$





from all interferograms (Berardino et al., 2002). In our case, the discontinuity line is no longer observable after applying the stacking technique (as shown in Figure 5(c)), thus having minimal impact on the final $z_{water}$ estimates.

### 3.3 Topographic artifacts related to DEM-SAR pixel misregistration

An important finding of this study is that pixel misregistration between the DEM and a SAR image can lead to DEM-related errors in InSAR LOS measurements. As an example, Figure 10(a) shows an interferogram formed by SAR images acquired on

8 June 2008 and 30 July 2010, and Figure 10(b) shows an interferogram formed by SAR images acquired on 3 June 2006 and 27 July 2009. Because the long-term subsidence trend is negligible, similar early June to late July thaw subsidence patterns are present in these two interferograms. To quantify InSAR LOS errors in areas with larger percent slopes ($> 7.5\%$), we zoomed in to a hilly area near Imnavait Creek with a slope between 8.4% and 11.0%, and calculated the phase difference (a phase difference of $2\pi$ is equivalent to 12 cm LOS distance difference for L-band ALOS PALSAR data) between points $P_E$ on

the east-facing slope and $P_W$ on the west-facing slopes across a hill ridge. Because water flows away from ridges with very small catchment areas, we expect to observe minimal freeze-thaw deformation on either side of the dry hill ridge. However, the phase difference between $P_E$ and $P_W$ is 1.23 rad (an equivalent LOS deformation error of 2.3 cm) for the interferogram that spans 8 June 2008 and 30 July 2010 (Figure 10(d)), and 0.92 rad (an equivalent LOS deformation error of 1.7 cm) for the interferogram that spans 3 June 2006 and 27 July 2009 (Figure 10(e)). Although the perpendicular baselines of these two

interferograms are similar ($\sim$ 1500 meters), the observed errors are different. These artifacts were observed in many Toolik ALOS interferograms across ridges (Figure 11). These phase artifacts are most noticeable in interferograms formed using one of the three SAR images (acquired on 8 June 2008, 24 October 2008, and 27 October 2009), which likely suffer more severe pixel misregistration errors than other SAR scenes. By contrast, an error in the DEM dataset itself can lead to LOS errors that are linearly proportional to the perpendicular baseline ($B_{perp}$) in all interferograms (Section 3.2), while long-term deformation

trend signals are proportional to temporal baselines (e.g., related to various slope processes as discussed in (Dini et al., 2019)).

To confirm that the observed InSAR phase errors between east-facing and west-facing slopes are indeed associated with DEM-SAR misregistration, the Kuparuk River watershed DEM data (Nolan, 2003a) were shifted to the east by 1 pixel ($\sim$ 12 m). The difference between the original and shifted DEM was used as an approximation of the DEM error ($\delta$) caused by 1-pixel misregistration to the east as described in Equation (7). In this case, a positive DEM error on the east-facing slope with

respect to the west-facing slope was observed (Figure 12 (a)). Similarly, a negative DEM error was observed on the east-facing slope with respect to the west-facing slope when the original DEM was shifted by 1 pixel to the west (Figure 12 (b)). When the DEM-SAR misregistration is in the north-south direction, DEM errors on the north-facing slope with respect to the south-facing slope were observed (Figure 12 (c) and (d)). Furthermore, at a given location, DEM errors increase as the amount of pixel misregistration increases to 2 pixels (Figure 12 (e)-(h)). Finally, the simulated DEM errors due to pixel misregistration

were compared to real LOS InSAR observations over the same region. Because ridges in the zoomed-in Imnavait Creek area are mainly along the north-south direction, the observed LOS error patterns in real ALOS Toolik interferograms (Figure 11) are most visible on the east-facing and west-facing slopes, which closely resemble DEM error patterns as shown in Figure 12 (a).





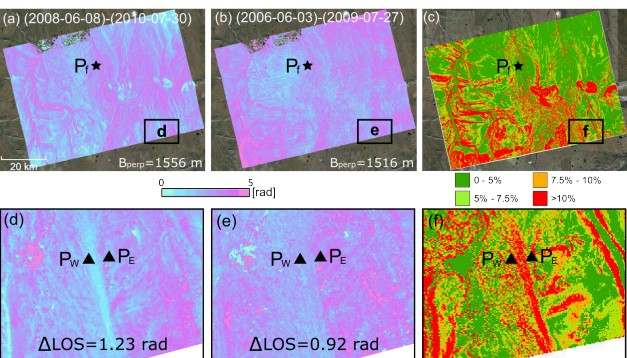

**Figure 10.** (Top) (a) An L-band ALOS interferogram (path 255 frame 1370) that spans 8 June 2008 and 30 July 2010. (b) An L-band ALOS interferogram (path 255 frame 1370) that spans 3 June 2006 and 27 July 2009. (c) A map of the percent slope in the study area. The black box outlines the zoomed-in area. Point $P_f$ ("f" stands for "flat") marks the location of a flat region analyzed in Figure 14. (Bottom) The InSAR phase measurement over the zoomed-in region outlined with the black box. $P_E$ and $P_W$ are on the east-facing and west-facing slopes. The phase difference between $P_E$ and $P_W$ is 1.23 rad in (d) and 0.92 rad in (e). A phase difference of $2\pi$ is equivalent to a 12 cm LOS error for L-band ALOS PALSAR data. (f) A map of the percent slope in the zoomed-in area.

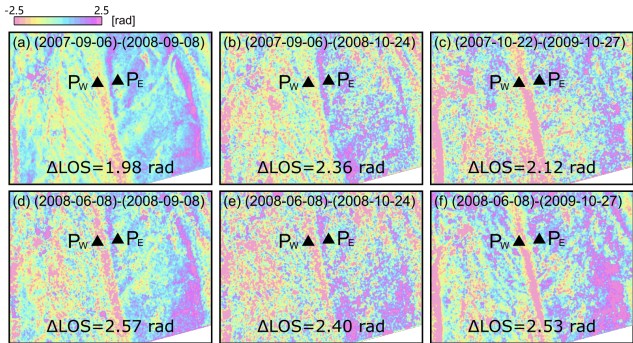

**Figure 11.** InSAR phase measurements over the zoomed-in region outlined with the black box in Figure 10 for interferograms with large pixel misregistration errors. $P_E$ and $P_W$ are on the east-facing and west-facing slopes. Here all interferograms were referenced to a local reference point near $P_E$.





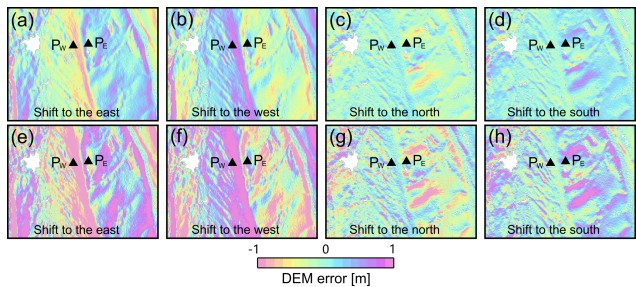

**Figure 12.** Estimated DEM errors (in m) introduced by 1 pixel shift ($\sim$ 12 m) to the (a) east, (b) west, (c) north, and (d) south, and shifting the DEM for 2 pixels ($\sim$ 24 m) to the (e) east, (f) west, (g) north, and (h) south. At a given pixel location, larger pixel misregistration leads to larger DEM errors. For fixed pixel misregistration, pixels with larger slopes show larger DEM errors. The study area is the same as in Figure 10, zoomed-in panels. The green color means the error is negligible. Purple and pink colors indicate positive and negative errors, respectively.

The land-surface slope is another factor that could affect the magnitude of DEM errors $\delta$ at different pixels. We classified
radar pixels into four groups based on their percent slope. For each group, phase errors associated with pixel-mismatching (1 radian phase error is equivalent to 1.9 cm LOS error) were calculated for the case that the location of the ridge is off by 1 pixel ($\sim$ 12 m) to the east and a perpendicular baseline of 5104 m. This scenario can be considered as the error upper bound because (1) the perpendicular baselines of L-band ALOS Toolik data are typically less than 5104 meters, and (2) the amount of pixel misregistration in the standard InSAR processing software packages is on the order of sub-pixels. We found that topographic
artifacts associated with DEM-SAR pixel misregistration are most noticeable in areas with a slope larger than 10%, and the majority of the surface area with a low slope (0-5%) show negligible phase errors due to DEM-SAR pixel misregistration (Figure 13). To further illustrate this, Figure 14 shows the amount of the LOS errors (in cm) in all interferograms at a steep area and a flat area. Up to $\sim$ 6 cm LOS errors associated with pixel misregistration were observed in the steep area, while $<$ 1 cm LOS errors were observed in the flat area.

In summary, we found that (1) the DEM error $\delta$ increases as the amount of pixel misregistration increases for a given pixel location (Figure 12); (2) the DEM error $\delta$ increases with local slopes at different pixel locations for the same amount of pixel misregistration (Figure 13 and Figure 14); and (3) the relationship between the LOS error due to $\delta$ and the perpendicular baseline is non-linear. We emphasize that both the amount of pixel misregistration and the slope influence the DEM error $\delta$. For example, $\delta$ equals 0 if there is no pixel misregistration. At a given pixel location, $\delta$ increases as the amount of pixel
misregistration increases. For a fixed amount of pixel misregistration, $\delta$ increases as the slope increases at different pixel locations. This means that the perpendicular baseline is not the only factor that controls the observed DEM artifacts in InSAR LOS measurements $\Delta d_{dem}$. It is difficult to fully correct the pixel misregistration because SAR images and DEM data were acquired from sensors with different spatial resolutions and imaging geometries. For example, the generation of the ArcticDEM using multiple imagery data acquired at different times can introduce distortions, which makes it challenging to precisely
quantify the propagation of this effect in the misregistration. Additionally, pixel misalignment could also be influenced by





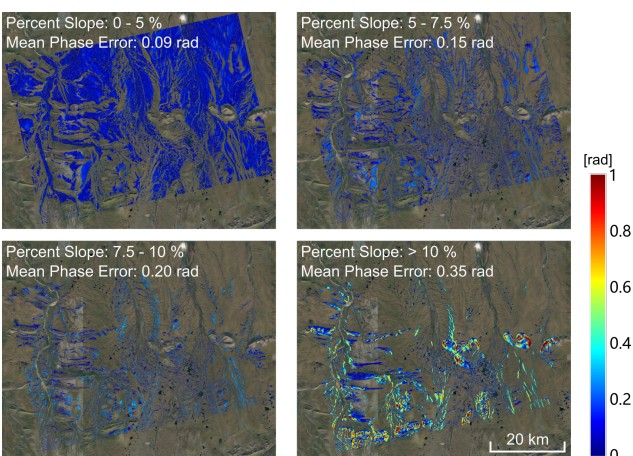

**Figure 13.** Estimated InSAR phase errors (in radians) at pixels with different slopes (0-5%, 5-7.5%, 7.5-10%, and >10%). Here DEM errors are introduced by 1 pixel misregistration ($\sim$ 12 m) to the east as in Figure 12(a). InSAR phase errors were calculated based on an InSAR perpendicular baseline of 5104 m. An error of 1 radian equals an LOS error of 1.9 cm for L-band ALOS data.

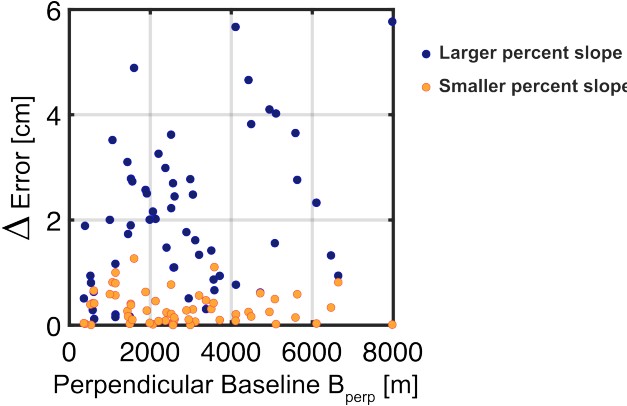

**Figure 14.** Line-of-sight (LOS) errors (in cm) of all interferogram pairs vs. perpendicular baseline (in m) at $P_E$-$P_W$ with percent slopes 8.4% to 11.0% (blue dots) and around $P_f$ with percent slopes 2.0% to 1.8% (orange dots). The location of these pixels is shown in Figure 10.





atmospheric distortions in optical imagery. Given that these pixel misregistration artifacts are mostly observed in a small subset of pixels with relatively large slope angles, we did not develop a misalignment correction algorithm in this study. Nonetheless, our approach provides a valuable way to identify and characterize pixel misregistration errors in the final LOS deformation estimates.

## 4    Conclusions

InSAR-estimated seasonal surface thaw subsidence measures the amount of water stored in the saturated soil active layer above permafrost, which can be used to constrain hydrologic models and water mass budgets. In the Toolik area, 95% of $z_{water}$ estimates range from 0 to 62 cm, and the spatial distribution of $z_{water}$ correlates with elevation and vegetation cover types. The amount of error in InSAR-estimated $z_{water}$ is linearly proportional to the error in InSAR LOS deformation measurements.

Although most InSAR measurement noises have been mitigated during the processing procedure, errors in the Arctic-DEM data and DEM-SAR misregistration can lead to visible InSAR LOS measurement errors. In the ALOS Toolik case, a 1-2 meter error in the Arctic-DEM data can lead to a LOS error larger than 1.5 cm when the perpendicular baseline is larger than 4000 m. Errors associated with the DEM-SAR misregistration are determined by the amount of pixel misregistration, the local slope, and InSAR perpendicular baselines. For the ALOS Toolik case, these pixel-mismatching artifacts are mainly

observed in regions with a steeper slope ($> 10\%$) in interferograms formed using a subset of SAR scenes with noticeable misregistration issues. Most pixels in our study area have percent slopes smaller than 5%, and the LOS measurement error is generally smaller than 1 cm. As the landscape near Toolik Lake on the North Slope of Alaska transitions from hilly terrain to the south to flat plains to the north, DEM-SAR misregistration no longer produces visible phase artifacts in InSAR LOS observations. Our study shows that InSAR is an effective and powerful technique for accurately monitoring the status of and

changes in hydrological characteristics in active-layer soils above continuous permafrost. InSAR estimates of soil water depth are statistically consistent with in situ observations, and the advantages of InSAR estimates include broader spatial coverage, higher spatial resolution, and the ability to map spatial patterns.

*Data availability.* ALOS PALSAR data were downloaded from the Alaska Satellite Facility at https://asf.alaska.edu/asfsardaac/. Arctic-DEM data were provided by the Polar Geospatial Center at https://www.pgc.umn.edu/data/arcticdem/. Kuparuk River watershed DEM data
were obtained at https://toolik.alaska.edu/gis/data/index.php. Toolik in situ soil measurements collected in the 2018 and 2019 summer field campaigns can be accessed from O'Connor et al. (2020).

## Appendix A:  Comparison between InSAR and Other Satellite Remote Sensing Techniques for Studying Water in the Active Layer

Observations from the Gravity Recovery and Climate Experiment (GRACE) mission have been used to estimate changes in

water mass within permafrost regions with a grid size of 1 arc degree, approximately 111 km. However, the spatial resolution

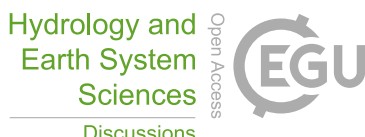

is too coarse for applications in understanding water flow in the active layer and for constraining most and especially detailed hydrologic models. Meanwhile, instead of estimating the amount of water stored in the active layer, the mass change observed by GRACE is mainly caused by mass loading by snow accumulation in winter and mass unloading by runoff in spring–summer (Muskett and Romanovsky, 2009).

In recent years, a spaceborne GNSS-R mission, the Cyclone Global Navigation Satellite System (CYGNSS) mission, has also been applied to study the freeze-thaw process in permafrost regions (Wu et al., 2020; Carreno-Luengo and Ruf, 2022). CYGNSS focuses on detecting the freeze-thaw state of the surface soil on top of the permafrost by monitoring changes in the dielectric constant. A regional map of the freeze-thaw state is derived by comparing the measured reflectivity with reflectivity measurements corresponding to frozen and thawed reference states. According to Carreno-Luengo and Ruf (2021), the detected

freeze-thaw state could be sensitive to properties of the top soil layer (0-7 cm), such as soil temperature and soil moisture content (SMC). However, according to our field data collected at $\sim$ 200 sites (marked in Figure 1 in the main paper), the active layer thickness in our study area has a mean of 56 cm with a quartile range of 44-67 cm, much thicker than the top soil layer (0-7 cm). Therefore, it is hard to infer soil water storage in the entire active layer using the information of SMC in the top soil layer. Moreover, in Carreno-Luengo and Ruf (2022), the authors showed that the SMC of the top soil layer actually

does not impact the freeze-thaw state results from CYGNSS data. Therefore, there does not exist a strong relationship between CYGNSS-detected freeze-thaw states and SMC in the top soil layer. To conclude, CYGNSS observations could be used to infer soil properties in the top soil layer, but monitoring soil water equivalent depth in the entire active layer would be difficult.

*Author contributions.* Y.W. was responsible for writing the original draft, while the review and editing process involved contributions from Y.W., J.C., M.B.C., and G.W.K. Visualization and validation was performed by Y.W. and J.C. The project was supervised by J.C., M.B.C.,

and G.W.K., with resources provided by J.C., M.B.C., and G.W.K. Project administration was handled by G.W.K. The methodology was developed collaboratively by Y.W., J.C., M.B.C., and G.W.K. The investigation was conducted by Y.W., who also carried out the formal analysis and data curation. Funding acquisition was undertaken by Y.W., J.C., and G.W.K. Finally, conceptualization was jointly accomplished by Y.W., J.C., M.B.C., and G.W.K.

*Competing interests.* The authors declare no competing interests.

*Acknowledgements.* This study was funded by the NASA Terrestrial Hydrology Program (80NSSC18K0983) and FINESST Program (80NSSC20K1622), with additional support from the National Science Foundation (ARC-1204220, DEB-1026843, 1637459, and 0639805, 2220863, and 2224743, PLR-1504006, and OPP-1107593, 1936759).



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





**Table A1.** Synthetic Aperture Radar scenes used in the study

| Date | Orbit | Path | Frame | Date | Orbit | Path | Frame |
|---|---|---|---|---|---|---|---|
| 2006/06/03 | 01900 | 255 | 1370 | 2006/06/03 | 01900 | 255 | 1380 |
| 2006/10/19 | 03913 | 255 | 1370 | 2006/10/19 | 03913 | 255 | 1380 |
| 2007/09/06 | 08610 | 255 | 1370 | 2007/09/06 | 08610 | 255 | 1380 |
| 2007/10/22 | 09281 | 255 | 1370 | 2007/10/22 | 09281 | 255 | 1380 |
| 2008/06/08 | 12636 | 255 | 1370 | 2008/06/08 | 12636 | 255 | 1380 |
| 2008/09/08 | 13978 | 255 | 1370 | 2008/10/24 | 14649 | 255 | 1380 |
| 2008/10/24 | 14649 | 255 | 1370 | 2009/07/27 | 18675 | 255 | 1380 |
| 2009/07/27 | 18675 | 255 | 1370 | 2009/09/11 | 19346 | 255 | 1380 |
| 2009/09/11 | 19346 | 255 | 1370 | 2009/10/27 | 20017 | 255 | 1380 |
| 2009/10/27 | 20017 | 255 | 1370 | 2010/06/14 | 23372 | 255 | 1380 |
| 2010/06/14 | 23372 | 255 | 1370 | 2010/07/30 | 24043 | 255 | 1380 |
| 2010/07/30 | 24043 | 255 | 1370 | | | | |





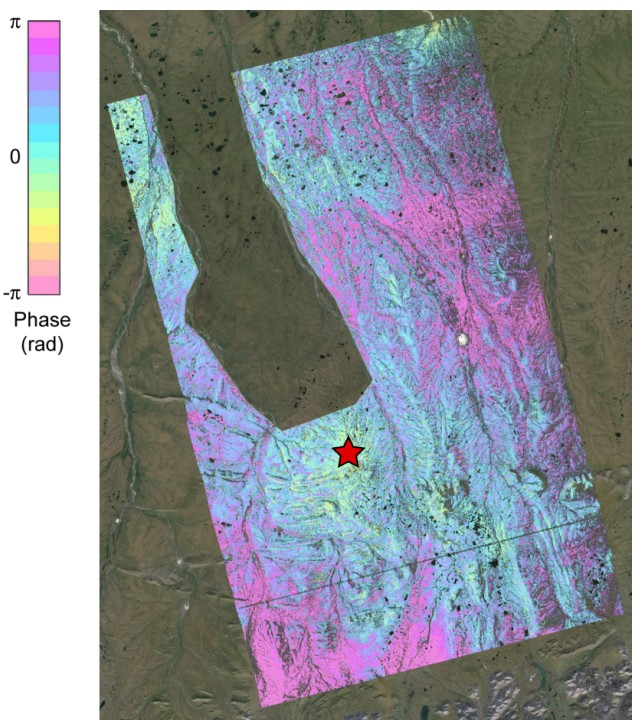

**Figure A1.** An L-band ALOS PALSAR interferogram (Path 255 Frame 1370-1380) that spans June 3, 2006 and July 27, 2009 over the study area. A phase cycle ($2\pi$) equals 12 cm radar Line-Of-Sight (LOS) deformation. Subsidence leads to positive LOS deformation (pink). The same reference point at 68.83° N, 150.23° W were used for both frames, as marked by the red star.





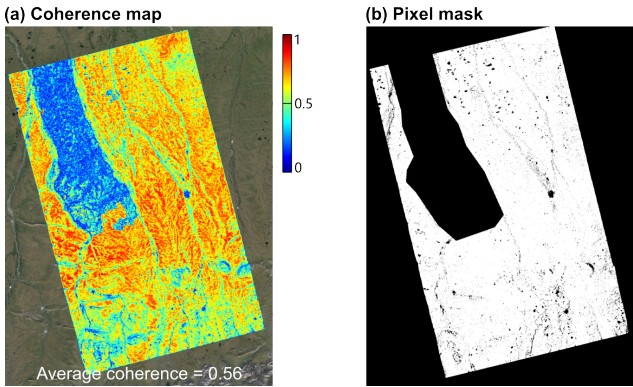

**Figure B1.** (a) A map of the average phase coherence of all interferograms used for estimating seasonal thaw subsidence. Low phase coherence is observed over water bodies and regions burned by the 2007 Anaktuvuk River fire. (b) The pixel mask used in this study. The black color indicates pixels that have been masked out or are outside the study area. This mask excludes any pixels with low amplitude and low phase coherence $< 0.2$ (e.g. water bodies and the area affected by the 2007 Anatuvuk River fire).

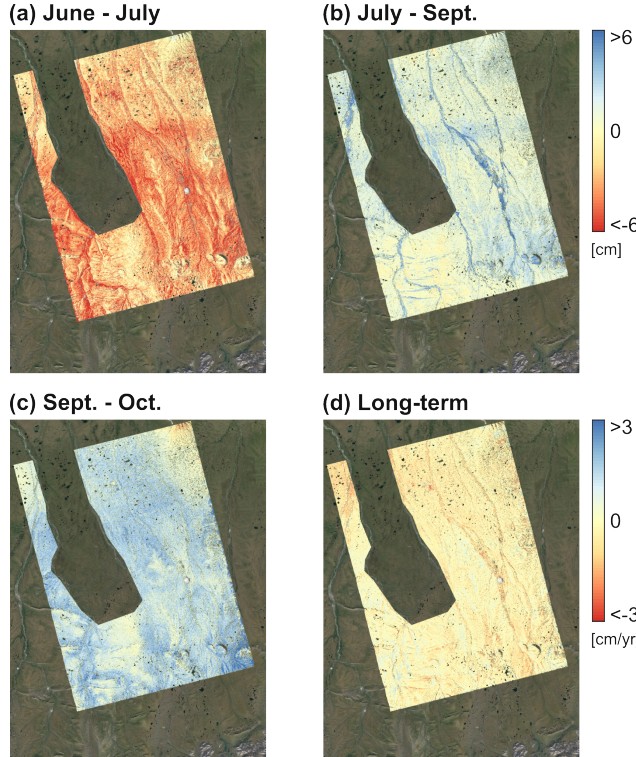

**Figure C1.** Average seasonal surface deformation (in cm) between (a) early June and late July, (b) late July and early September, and (c) early September and late October derived from ALOS PALSAR Path 255 Frame 1380 InSAR observations. (d) Average long-term surface deformation trend (cm/yr) between 2006 and 2010 derived from ALOS PALSAR Path 255 Frame 1380 InSAR observations. Here red means subsidence, yellow means no significant deformation, and blue means uplift. The area affected by the 2007 Anaktuvuk River fire, along with water bodies, has been masked out based on InSAR phase coherence. Toolik Field Station in-situ data suggest that the air temperature fluctuated around or below freezing in early September during the ALOS PALSAR data acquisition times (at $\sim$ 12 am local time). In this scenario, ice can be formed at the top of the soil, which leads to frost heave in saturated soils (Chen et al., 2020)

.





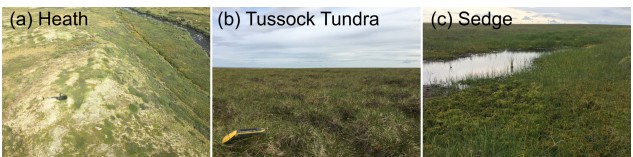

**Figure D1.** Field photos of (a) heath land cover, (b) tussock tundra land cover, (c) wet sedge land cover (Adapted from Chen et al. (2020)).





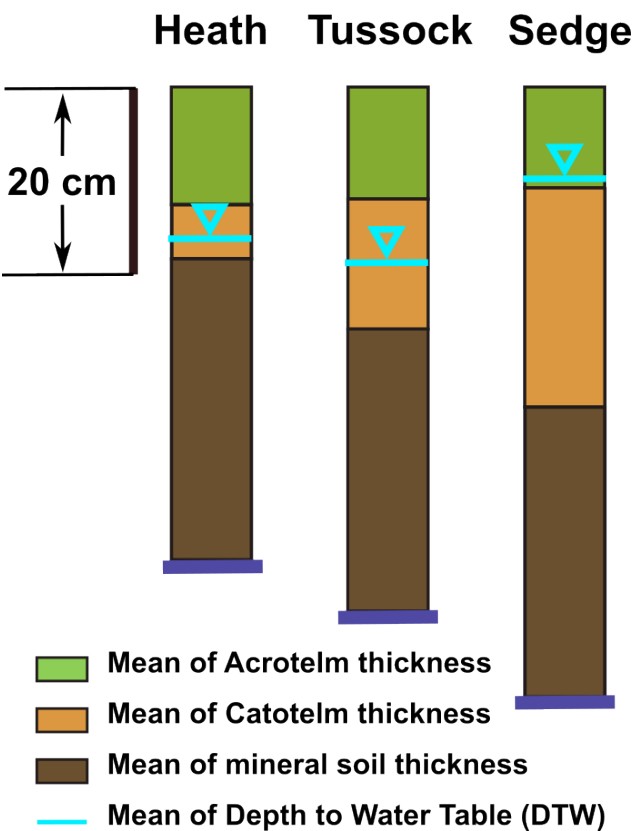

**Figure E1.** Soil stratigraphy under different land covers near Toolik Lake area as derived from field measurements collected at sites marked in Figure 1 in the main paper (Adapted from Chen et al. (2020)).