# Peer review of "High-resolution InSAR Regional Soil Water Storage Mapping Above Permafrost"

_Hydrology and Earth System Sciences, 2024_

## Author Comment (AC1)

We appreciate the time that the reviewer put into this paper. Below, we provide detailed explanations of how we plan to revise the manuscript. The reviewer's comments are in *blue italics*, our responses are in black, and relevant text from the manuscript, whether newly added or edited, is **bold**. We are grateful for the reviewer's insightful feedback.

*This manuscript presents a methodology for estimating the volumetric storage of soil water within the active layer of periglacial landscapes using InSAR observations. The method from a prior proof of concept publication is re-introduced and applied to a larger area of undisturbed tundra in the vicinity of Toolik Field Station on the Alaskan North Slope using L-band ALOS data acquired between 2006 and 2010. The InSAR-estimated equivalent water depth estimates are compared with in situ estimates, showing strong overall agreement. Lastly, there is a discussion of sources of interferometric error due to DEM errors and SAR image to DEM pixel misregistration.*
*Overall, I think that this manuscript presents several relevant contributions. However, I would first recommend several revisions before the manuscript can be considered for publication. I have a detailed, point-by-point list of specific comments below, but these comments generally fall under a few broader thematic points. These points are:*
1. *The manuscript would benefit from a more precise description of what the Zwater parameter physically corresponds to. From my reading of prior work, the Zwater parameter is the equivalent water depth of all pore-bound water undergoing seasonal freeze/thaw within the active layer. However, the Zwater parameter is variously referred to in slightly different ways throughout the manuscript, which could raise questions from readers as to what exactly Zwater physically corresponds (i.e., is this the equivalent water depth, or the thickness of the active layer that is fully saturated?) I have a few specific suggestions to address this.*
   1) *First, I would recommend including a physical cartoon or schematic that unambiguously illustrates the Zwater parameter. For example, Figure 2 could be modified by including two additional cartoon cross-sections next to the original soil profiles, with the total porebound water separated from the soil matrix into a single homogenous column of pure water, with the depth of this water column labeled Zwater.*
   2) *I recommend choosing a standard way to describe Zwater and being consistent throughout the manuscript. At various times Zwater is referred to as 'saturated active layer soil water storage', 'saturated soil water thickness', 'soil water equivalent depth', 'soil water depth in the saturated active layer', 'soil water column that experiences the ice-to-water phase change in the saturated active layer (denoted as zwater)', 'active layer water storage' and 'saturated active layer soil water column'.*

**Response:** In both Chen et al. (2020) and the current paper, we stated that InSAR measures the total soil water storage in the saturated active layer. InSAR thaw subsidence is not sensitive to soil water stored above the water table (tension water or unsaturated-zone pore water, Figure 2). Because the porosity in the organic soil layers is high (~0.78-0.98), water in the unsaturated zone can expand to fill the empty pore space (soil atmosphere) during freezing without contributing to surface deformation. We acknowledge that Chen et al. (2020) did not explicitly discuss the relationship between InSAR thaw subsidence observations and the total soil water stored in the saturated active layer (that was not the main point of that paper). An important contribution of this current paper is to establish the conceptual model that explains what InSAR thaw subsidence over permafrost terrain measures as described in Section 2.1.

Following the reviewer's comment, we updated Figure 2 as:

[Figure]

The figure caption of Figure 2 was updated as:

**Figure 2. Conceptual diagram of how the depth of saturated water ($Z_{water}$) affects soil surface deformation. The soil is divided into three zones: (1) the surface active layer unsaturated zone (*thickness of u*), which contains soil particles (dark circles), soil atmosphere (white open space), and it may contain tension (capillary) water (shown in light blue in column (B)); (2) the water-saturated active layer (*thickness of s*), the upper surface of which defines the water table and the lower surface defines the ice table, which separates the thawed active layer from frozen ground (or the permafrost layer at maximum annual thaw depth); and (3) the permafrost (*thickness of p*), which may or may not be saturated with water as ice. In column (A) the $Z_{water}$ thickness represents the total amount of water in the saturated active layer. In (B), the saturated active layer is the same thickness as in column (A), but there is tension water in the upper unsaturated zone. However, $Z_{water}$ thickness is the same in columns (A) and (B), independent of differences in tension water. The reason is shown in (C), where the entire soil column has now frozen. The saturated water freezes and the expansion heaves the soil column above and the ground surface (*thickness s + 0.09\*$Z_{water}$*), while the tension water freezes but expands into pores containing soil atmosphere and thus does not contribute to deformation of the ground surface (*thickness u does not change*).**

In the revised paper, we improved the clarity of Section 2.1, and we used the term 'saturated soil water storage in the active layer' or 'the soil water stored in the saturated active layer' consistently throughout the manuscript.

> 2. *As currently written, the section of the manuscript on Zwater estimation and the section on DEM errors feel rather disjoint from each other. I think that the manuscript would benefit from a tighter coupling of these two elements, as well as a more thorough and explicit*

*discussion of the novel advancements made in this manuscript on top of the Chen et al. 2020 proof of concept manuscript.*

    *1) The manuscript would benefit from a more explicit framing of the novel contributions that this manuscript introduces to the field over the prior Chen et al. 2020 study. The discrete dem error due to pixel misregistration analysis is one such novel contribution. However, I would also highlight any advancements made in the Zwater estimation process introduced.*

    *2) Currently, the Zwater and DEM error sections read as pretty disjoint from each other; these sections could potentially be submitted separately as two stand-alone papers; or, perhaps more discussion linking them could be included in the revised manuscript. For example, to what degree to observed DEM errors propagate into an effective uncertainty for Zwater estimates? Bridging these two sections together through an uncertainty propagation analysis would be one such way to strengthen the cohesiveness of the manuscript.*

**Response:** Following the reviewer's recommendation, we rewrote the last paragraph of the introduction to better illustrate the new contribution made in this paper on top of the Chen et al., (2020) proof of concept paper:

**"Our recent study found that the signal amplitude of the seasonal thaw subsidence is proportional to the amount of water stored in the saturated active layer at the end of a thaw season (Chen et al., 2020). In that paper, we further established a conceptual model that relates InSAR seasonal thaw subsidence observations to the amount of water in the saturated active layer. In the current paper we advance InSAR techniques for the high-resolution mapping of water storage above-permafrost. To demonstrate this, we mapped soil water stored in the saturated active layer using ALOS PALSAR data over a much larger area in the Arctic Foothills than in Chen et al. (2020). We validated the InSAR results using in-situ soil measurements collected at more than 200 remote sites within ~ 100 km of the Toolik Field Station as well as optical imagery and land cover maps. Our results show that InSAR soil water storage estimates derived from two separate satellite frames are consistent with in-situ observations under different vegetation covers. An important contribution of this work is on uncertainty quantification. We determine the primary error sources in Toolik ALOS PALSAR Line-Of-Sight (LOS) measurements, and we discuss how errors in InSAR LOW measurements can be linearly related to errors in soil water storage estimates."**.

We note that the uncertainty propagation analysis as suggested by the reviewer is covered in Section 2.2 following Equation (4), and InSAR error sources are discussed in Section 2.3. The results on the uncertainty analysis are presented in Section 3.2 and Section 3.3.

*Individual comments are organized by section below:*

*Introduction:*
*Page 1, line 22: "Whether the carbon held by the active layer soils will be transformed to carbon dioxide or methane (a more powerful greenhouse gas), or whether it will flow towards rivers and lakes as dissolved carbon in groundwater, depends largely on the wetness or dryness of the active layer (i.e., how much water is stored)." I would recommend including a citation or reference to a few relevant papers that support this statement, as it is a central point that underlines much of the scientific justification for this manuscript.*

**Response:** We included two new citations in the sentence on Line 22-24:

**"Whether the carbon held by the active layer soils will be transformed to carbon dioxide or methane (a more powerful greenhouse gas), or whether it will flow towards rivers and lakes as dissolved carbon in groundwater, depends largely on the wetness or dryness (i.e., how much water is stored) of the active layer (Bond-Lamberty et al., 2016; Taylor et al., 2021)."**

**References:**

Bond-Lamberty, B., Smith, A. P., & Bailey, V. (2016). Temperature and moisture effects on greenhouse gas emissions from deep active-layer boreal soils. Biogeosciences, 13(24), 6669–6681. https://doi.org/10.5194/bg-13-6669-2016

Taylor, M. A., Celis, G., Ledman, J. D., Mauritz, M., Natali, S. M., Pegoraro, E. -F., Schädel, C., & Schuur, E. A. (2021). Experimental soil warming and permafrost thaw increase $CH_4$ emissions in an upland tundra ecosystem. Journal of Geophysical Research: Biogeosciences, 126(11). https://doi.org/10.1029/2021jg006376

*Page 2 line 37: "Because ice density is less than water density (and thus ice volume is greater than water volume), the land surface subsides as the active layer thaws from winter to summer (Liu et al., 2010)" I would recommend being explicit here and stating that the amount of surface subsidence depends upon the overall volumetric water content of the thawing permafrost, as this further motivates the proposed methodology. One second thought, this may not be necessary to state here, as you later state it on line 44.*

**Response:** Yes, we did state this on line 44, and as noted earlier, we rewrote the last paragraph of the introduction to better illustrate the new contribution made in this paper on top of the Chen et al., (2020) proof of concept paper.

*Methods:*
*An important point to raise is that, in addition to assuming stationary thaw conditions from year to year, the interannual stacking method also implicitly assumes no variations in excess ground ice content from year to year. While I think that this is a justifiable assumption, it might be worth explicitly mentioning this, and discussing recent work that has demonstrated that InSAR is sensitive to interannual variations in excess ground ice formation and melting: https://doi.org/10.5194/tc-15-2041-2021, https://doi.org/10.1029/2023WR035331*

**Response:** We agree with the reviewer that interannual variation in excess ground ice formation and melting are key processes that can lead to variations in observed seasonal thaw subsidence from year to year. The conceptual model described in Section 2.1 can be used to study inter-annual variation in excess ground ice formation and melting, when high quality InSAR thaw subsidence observations of a single thaw subsidence are available. We clarified at the end of Section 3.1:

**"Due to the limited ALOS PALSAR data availability, the investigation of inter-annual variability of InSAR thaw subsidence patterns is outside the scope of this work. Future work can focus on studying how the signal magnitude of seasonal thaw subsidence changes over multiple years using Sentinel-1 data collected with 6-12 day revisit cycles (Zwieback and Meyer, 2021; Zwieback et al., 2024)".**

**References:**

Zwieback, S., & Meyer, F. J. (2021). Top-of-permafrost ground ice indicated by remotely sensed late-season subsidence. The Cryosphere, 15(4), 2041–2055. https://doi.org/10.5194/tc-15-2041-2021

Zwieback, S., Iwahana, G., Sakhalkar, S., Biessel, R., Taylor, S., & Meyer, F. J. (2024). Excess ground ice profiles in continuous permafrost mapped from InSAR subsidence. Water Resources Research, 60(2). https://doi.org/10.1029/2023wr035331

*Section 2.3:*
*-What are typical pixel misregistration values for ALOS? They are surely processor dependent, and the InSAR processor used is not explicitly mentioned. However, I imagine they are still relatively small, no more than 1 or 2 pixels in any direction, and usually sub-pixel.*

**Response:** We clarified in Section 2.3:
**"In this study, we employed the same image co-registration routine as the standard InSAR processing software such as the InSAR Scientific Computing Environment (ISCE) (Rosen et al., 2012) or GMTSAR (Sandwell et al., 2011). The 2-D cross-correlation method for image alignment can achieve sub-pixel accuracy in most cases. However, the alignment can be worse than 1 pixel, because SAR images and DEM data were acquired from sensors with different spatial resolutions and imaging geometries".**

**References:**
Rosen, P. A., Gurrola, E., Sacco, G. F., & Zebker, H. (2012). The InSAR Scientific Computing Environment. In *EUSAR 2012; 9th European Conference on Synthetic Aperture Radar*, pp. 730–733.
Sandwell, D. R.Mellors X.Tong M.Wei and P.Wessel (2011), Open Radar Interferometry Software for Mapping Surface Deformation, *Eos Trans. AGU*, 92(28), 234.

*Page 10 line 252: This is a minor point, but the comparison between the InSAR pixel (10-100 m) and field measurement (30x30 cm^2 area plot) is a 'linear to area' comparison.*
**Response:** We updated the text: "*A pixel in an InSAR-derived deformation map is ~ 100-by-100 meter, while field measurements were collected at sites with size ~1000 cm$^2$ (30-by-30 cm plots)*". Note that we are comparing a sampling area to a pixel area.

*Page 11 line 260: I suggest removing, or restating 'waviness of the ice-table', as it is not precise.*
**Response:** we removed 'waviness of the ice-table'.

*Page 11 line 260: "we found that the fitted PDFs stayed mostly the same" This statement is also imprecise. Can you quantify what 'mostly the same' means?*
**Response:** Below we include the PDF fitting results as presented in Chen et al., (2020). We clarified that:
**"To reduce estimation bias, we targeted specific vegetation cover types and soil layers needing larger sample sizes over time to improve statistical robustness. The PDF fitting results changed very little after a second year of sampling, indicating that the sample size in this study is sufficiently large to capture the statistical characteristics of soil properties".**
We inspected the histogram fitting based on visual inspection. Additionally, we calculated the Bhattacharyya distance between the fitted PDFs. The calculated Bhattacharyya distance (Bhattacharyya, 1946) between the 2018 and 2019 fitted PDFs ranged from 0.07 to 0.12, suggesting that the sample size in this study was large enough to reliably represent the soil properties at the regional scale."

**Reference:**

A. Bhattacharyya, "On a measure of divergence between two statistical populations defined by probability distributions" Bull. Calcutta Math. Soc. , 35 (1943) pp. 99–109

[Figure]

*Results and Discussion:*
*Section 3.3:*
*Page 18 line 387: Why do these three scenes likely exhibit more severe pixel misregistration errors compared to the other scenes?*
**Response:** These scenes tend to have a relatively large spatial baseline with respect to the reference orbit. In this case, this leads to more noticeable image distortion, which makes it more difficult to track the pixel offset.

*Can an alternative (or compounding) interpretation for the observed phase difference between east and west facing slopes be due to viewing geometry rather than DEM misregistration? Comparing to a descending path frame track over the same area would shed light on this, and allow the authors to rule out a difference in the projection of downslope deformation (e.g., solifluction) onto the LOS vector vs. a pixel misregistration issue.*
**Response:** Unfortunately, only ascending ALOS data are available in this area. Nonetheless, we predicted the phase errors due to DEM mis-alignment (Figure 12) based on Equations (6)-(7), which look very similar to actual phase observations from multiple interferograms (Figure 11). We evaluated the estimated InSAR phase errors due to DEM-misalignment in magnitude and spatial distribution (Figure 13). At the same location, this error increases with InSAR perpendicular baselines (Figure 14; blue dots), a key feature of DEM-related InSAR phase errors. By contrast, deformation signals related to solifluction processes are not controlled by InSAR perpendicular baselines, and they should not resemble the simulated DEM mis-alignment patterns (Figure 12) so closely.

*Conclusions:*
*Page 22, line 431: "InSAR-estimated seasonal surface thaw subsidence measures the amount of water stored in the saturated soil active layer above permafrost, which can be used to constrain hydrologic models and water mass budgets." Rather than saying InSAR thaw subsidence measures*

*water storage, I might suggest instead something like 'is sensitive to' or 'is related to', as this is not a direct measurement of soil water storage, but rather a model-based estimation.*
**Response:** We changed "*measures*" to "*can be related to*" as suggested by the reviewer.

---

## Author Comment (AC2)

We appreciate the time that the reviewer put into this paper. Below, we provide detailed explanations of how we plan to revise the manuscript. The reviewer's comments are in *blue italics*, our responses are in black, and relevant text from the manuscript, whether newly added or edited, is **bold**. We are grateful for the reviewer's insightful feedback.

*This paper builds on previous work by the same research team (Chen et al., Remote Sensing of Environment, 2020), which utilized satellite radar interferometry (InSAR) to estimate frozen ground properties. The current study extends that work by focusing on total soil water storage within the entire active layer column, an important hydrological property that remains poorly constrained across vast permafrost regions.*

*Major comments:*
*The previous work by Chen et al. (2020) established a link between seasonal subsidence and active layer water storage. The primary advancements in this latest work include presenting water storage as a key result, incorporating InSAR data from an additional satellite frame to extend the study area, and identifying InSAR errors caused by DEM defects. While these contributions are valuable, the paper in its current form does not sufficiently emphasize the fundamental novelty of the methodology. The stacking approach and the scaling of seasonal subsidence by density difference to derive total water content remain the same as the previous work. If these methods are not fundamentally new, I recommend shortening the related methodological section and referring readers to Chen et al. (2020) for more details.*

**Response:** Following the reviewer's comment, we rewrote the last paragraph of the introduction to better illustrate the new contribution made in this paper compared to the Chen et al., (2020) proof of concept paper: **"Chen et al. (2020) found that the amplitude of the seasonal thaw subsidence is proportional to the amount of water stored in the saturated active layer at the end of a thaw season. This is consistent with findings from recent studies that InSAR-derived seasonal subsidence rates reflect spatial soil moisture patterns (Chen et al., 2022, 2023; Widhalm et al., 2024). In this paper, we further established a conceptual model that relates InSAR seasonal thaw subsidence observations to soil water storage in the saturated active layer. Our goal is to advance InSAR techniques for the high-resolution mapping of water storage above-permafrost. To demonstrate this, we mapped soil water stored in the saturated active layer using ALOS PALSAR data over a much larger area in the Arctic Foothills than used in Chen et al. (2020). We validated the InSAR results using in-situ soil measurements collected at more than 200 remote sites within ~ 100 km of the Toolik Field Station as well as optical imagery and land cover maps. Our results show that InSAR soil water storage estimates derived from two separate satellite frames are consistent with in-situ observations under different vegetation covers. An important new contribution of this work is on uncertainty quantification. We determine the primary error sources in Toolik ALOS PALSAR Line-Of-Sight (LOS) measurements, and we discuss how errors in InSAR LOW measurements can be linearly related to errors in soil water storage estimates".**

We summarized our InSAR processing strategy and cited Chen et al. (2020) in the original draft. However, one reviewer noted earlier: "*although the authors clearly indicated that the same procedure proposed by Chen et al. 2020 is used, they should provide more information/details regarding the InSAR procedure/strategy*". This is why we expanded Section 2.2 to better describe the InSAR processing method, which makes the paper self-contained and easier to follow. The uncertainty propagation analysis is covered in Section 2.2 following Equation (4), and InSAR major error sources are discussed in Section 2.3. The results on the uncertainty analysis are presented in Section 3.2 and Section 3.3.

*The paper highlights DEM error issues that cause noticeable artifacts in individual interferograms at fine, local scales. While these issues are worth noting, the authors did not correct these errors in their InSAR processing. Moreover, it appears that these errors do not affect the final water storage estimates after stacking multiple interferograms (e.g., Figure 5c).*

**Response:**  The reviewer is correct, the DEM coregistration errors only affect areas of high slope angle, and we did not individually correct each pixel for potential coregistration errors (as described fully below). An important contribution of this work is to quantify the uncertainty in the active layer soil water storage estimates from major InSAR measurement error sources (Section 2.3). This is an important step forward to enable accurate interpretation of InSAR observations over permafrost terrain, and scalability and uncertainty quantification are important in new remote sensing water storage retrieval algorithm development.

In Section 2.3, we conducted quantitative assessment of DEM coregistration errors, and discussed the results in Section 3.3 as follows: We predicted the phase errors due to DEM mis-alignment (Figure 12) based on Equation (6)-(7), and compared them to the phase observations from multiple interferograms (Figure 11). We evaluated the estimated InSAR phase errors due to DEM-misalignment in magnitude and spatial distribution (Figure 13). We show that this error increases with InSAR perpendicular baselines (Figure 14; blue dots), a key feature of DEM-related InSAR phase errors rather than other geophysical processes such as solifluction. We note that these DEM artifacts are not just visible in individual interferograms, but they are present in the InSAR thaw subsidence map (Figure 5a), mostly at pixels with large slope angles (Figure 13).

It is difficult to fully correct the pixel misregistration issues. In this study, we removed the topographic phase during interferogram formation using the Arctic DEM v3.0 data (Porter et al., 2018), which are widely used in the Arctic community because of its pan-arctic coverage and high quality (Tozer et al., 2019). We used the same image co-registration routine as the standard InSAR processing software such as GMTSAR and ISCE (Co-author J. Chen is on the GMTSAR developer team). The 2-D cross correlation method for image alignment can achieve ~ 0.1-pixel accuracy in the best-case scenario. However, the accuracy can be worse than 1 pixel because SAR images and DEM data were acquired from sensors with different spatial resolutions, imaging geometries, and uncertainties. Because the pixel offset between SAR and DEM images is not a constant, a manual adjustment would have to be performed at each individual pixel, which is not practically feasible. Based on the fact that these pixel misregistration artifacts are mostly observed in a small subset of pixels with relatively large slope angles (Figure 13), we conclude InSAR is a feasible technique for regional active layer water storage mapping over our study site. Future work may focus on the development of a misalignment correction algorithm.

*For the HESS readership, I suggest strengthening and elaborating more on the hydrological significance of this method and results. For instance, what are the advantages and limitations of estimating soil water content using this method? What new insights are gained from the estimated water storage in the context of permafrost hydrology?*

**Response:** This paper uses space geodetic observations to map water stored near the earth's surface. We stated in the abstract: "**The hydrology of thawing permafrost affects the fate of the vast amount of permafrost carbon due to its controls on waterlogging, redox status, and transport. However, regional mapping of soil water storage in the soil layer that experiences annual freeze-thaw cycles above permafrost, known as the active layer, remains a formidable challenge over remote arctic regions**". We further justified the scientific rationale of the work in the first paragraph of introduction. Existing InSAR permafrost studies tended to associate the magnitude of the InSAR-observed thaw subsidence with the active layer thickness. However, our recent work (Chen et al., 2020) shows that the amplitude of the maximum seasonal thaw subsidence is proportional to the amount of soil water that experiences the active layer freeze-thaw cycle (not necessarily the active layer thickness). This means that satellite remote sensing of surface deformation is a potential strategy for mapping water storage in the active layer with broad coverage and relatively high spatial resolution. In this paper, we further developed a conceptual model (Figure 2) that

relates InSAR seasonal thaw subsidence observations to soil water storage in the saturated active layer (Section 2.1). The resulting InSAR active layer water storage map will be an interest to many people who work in the field of permafrost hydrology research, including but not limited to the remote sensing scientists.

An important assumption we employ is that the observed InSAR deformation is associated with the active layer freeze-thaw processes. At the end of Section 2.1, we discussed various hydrological and geophysical processes that may lead to deformation in permafrost environment: "**We emphasize that many geophysical processes can lead to surface deformation in permafrost terrain detectable by InSAR (Zwieback et al., 2024b). For example, slope creep processes may produce long-term downward deformation trends in regions with large slope angles (Dini et al., 2019). Post-glacial rebound and tectonic motions typically vary at 100-km or larger spatial scales and can be considered as nearly spatially uniform over our study area (Liu et al., 2010; Stephenson et al., 2022). Given that InSAR measures relative deformation with respect to a local reference point, InSAR is only sensitive to spatially varied surface deformation over the study area. Hydrological loading and unloading can produce millimeter-level surface deformation signals (Liu et al., 2010), which is much smaller than centimeter-level freeze-thaw deformation. Furthermore, peat accumulation and erosion processes (Jones et al., 2017) can cause changes in surface scattering properties, which decorrelate radar phase measurements (Zebker and Villasenor, 1992). As a result, it is difficult to capture these processes using InSAR. In Section 2.2, we discuss how to extract long-term and seasonal deformation signals from InSAR observations. The magnitude and characteristics of deformation signals, combined with in-situ observations (Section 2.4), can be used to determine the primary geophysical processes that contribute to the observed deformation patterns**".

*The rigor of this work could be further improved by providing a quantification of the uncertainties associated with the water storage estimates and discussing the limitations of the method (again, in the context of permafrost hydrology). For example, the simple scaling does not account for unfrozen water, excess ice, or vertical moisture migration; the stacking-based InSAR processing might not be applicable in cases of significant inter-annual variations and linear secular trends; other InSAR errors such as tropospheric delay or phase artifacts due to soil moisture changes are not explicitly accounted, etc.*

**Response:** As noted in our response above, we discussed various geophysical processes that may lead to deformation in permafrost environment, and our method is based on the assumption that the active layer freeze-thaw process is the primary geophysical process that contributes to the the observed deformation. Given the characteristics of the observed deformation signals (Figure B3) and in-situ validation results, this assumption is valid over the Toolik area. As shown in Figure 2, water in the unsaturated zone (tension water) can expand to fill the empty pore space during freezing without contributing to surface deformation. In Section 2.1, we further emphasize that the InSAR method is not sensitive to water in the active layer that does not experience the annual freeze-thaw cycle (e.g., the runoff term Q in Equation 2). Because InSAR measures the total deformation over all depths, it is not sensitive to vertical moisture migration.

In terms of secular trends, in Section 2.2 we state that: "**We first solved for the long-term LOS deformation trend at a pixel of interest based on a stacking approach. That is, averaging all interferograms that contain minimal seasonal deformation signals (e.g., a July-to-July pair) and relatively large long-term signals (e.g., span multiple freeze-thaw cycles)**". We note that stacking methods have been used to solve for the linear deformation trend over multiple periods of time, and the results are comparable to the SBAS InSAR time series method (Staniewicz et al., 2020). At the end of Section 3.1, we further discussed the

limitation of the stacking method: **"Due to the limited ALOS PALSAR temporal sampling rate, the investigation of inter-annual variability of InSAR thaw subsidence patterns is outside the scope of this work. Future work can focus on studying how the signal magnitude of seasonal thaw subsidence changes over multiple years using Sentinel-1 data collected with 6-12 day revisit cycles (Zwieback and Meyer, 2021; Zwieback et al., 2024)".**

In terms of other InSAR errors, in Section 2.3 we discussed the major error sources including tropospheric noise. We included key references on InSAR tropospheric noise studies (Zebker et al., 1997; Emardson et al., 2003) to support our assumption that residual tropospheric noise level in individual interferograms is ~ 2 cm (after long-wavelength tropospheric noise was removed during the planar ramp removal process). We typically expect large tropospheric noise in hot and humid environments. Due to a cool and dry tundra climate over the Toolik area, we do not expect the tropospheric noise level to be substantially higher than the values reported in Emardson's 2003 South California study. We note that a thaw subsidence pattern similar to the final stacking solution was identified from all individual interferograms that span a common season, and the differences (a measure of noise terms) are on the order of centimeters. This is also consistent with the assumption that residual tropospheric noise level in individual interferograms is ~ 2 cm. Stacking reduces the impact of random noise by $\sqrt{N}$, where N is the number of independent SAR acquisitions. As a result, the turbulent random noise level can be reduced to less than 1 cm after stacking four interferograms formed from four SAR acquisitions. The change in soil moisture can lead to closure phase errors, which is typically much smaller than the tropospheric noise term in Equation (5).

Reference:
Staniewicz, S., Chen, J., Lee, H., Olson, J., Savvaidis, A., Reedy, R., Breton, C, Rathje, E., and Hennings, P. (2020). InSAR reveals complex surface deformation patterns over an 80,000 km2 oil-producing region in the Permian Basin. Geophysical Research Letters, 47, e2020GL090151. https://doi.org/10.1029/2020GL090151.

*Minor comments:*
*Section 1:*
*Some papers published in recent years have made similar attempts to estimate soil water content above permafrost using InSAR. Consider citing some of them and highlighting your contributions.*

- *Chen, J., Wu, T., Liu, L., Gong, W., Zwieback, S., Zou, D., Zhu, X., Hu, G., Du, E., Wu, X., Li, R., and Yang S. (2022), Increased water content in the active layer revealed by regional-scale InSAR and independent component analysis on the central Qinghai-Tibet Plateau, Geophysical Research Letters, 49, e2021GL097586, https://doi.org/10.1029/2021GL097586.*
- *Chen, R. H., Michaelides, R. J., Zhao, Y., Huang, L., Wig, E., Sullivan, T. D., Parsekian, A. D., Zebker, H. A., Moghaddam, M., and Schaefer, K. M. (2023), Permafrost Dynamics Observatory (PDO): 2. Joint Retrieval of Permafrost Active Layer Thickness and Soil Moisture From L-Band InSAR and P-Band PolSAR, Earth and Space Science, 10, e2022EA002453, https://doi.org/https://doi.org/10.1029/2022EA002453*
- *Widhalm et al. InSAR-derived seasonal subsidence reflects spatial soil moisture patterns in Arctic lowland permafrost regions, https://egusphere.copernicus.org/preprints/2024/egusphere-2024-2356 (Paper accepted for publication in The Cryosphere, title to be changed)*

*And two recent review papers for your reference.*

- *Zwieback, S., Liu, L., Rouyet, L., Short, N., and Strozzi, T. (2024), Advances in InSAR Analysis of Permafrost Terrain, Permafrost and Periglacial Processes, https://doi.org/10.1002/ppp.2248.*
- *Streletskiy, D., Maslakov, A., Grosse, G., Shiklomanov, N., Farquharson, L., Zwieback, S., Iwahana, I., Bartsch, A., Liu, L., Strozzi, T., Lee, H., and Debolskiy, M. (2025), Thawing*

*permafrost is subsiding in the Northern Hemisphere–review and perspectives, Environmental Research Letters, 20, 013006, https://doi.org/10.1088/1748-9326/ada2ff.*

**Response:** Thank you for these references, we now have them cited appropriately in our manuscript. As noted in our response to the major comments, we rewrote the introduction to better illustrate the new contribution made in this paper beyond that in Chen et al., (2020). We also discussed the assumptions and limitations of the method in Section 2.1. We cited recent work noted here in the revised paper.

*Section 2:*
*(Also relevant to section 3.2) Consider using alternative DEM such as the Copernicus GLO-30 Digital Elevation Model or the latest release of ArcticDEM (v4.1) to quantify and even reduce DEM errors.*

**Response:** We used both Kuparuk River River watershed DEM and Arctic DEM in the original InSAR analysis, and the resulting interferograms have comparable quality. We recently reprocessed the ALOS PALSAR interferograms using the updated ArcticDEM, and we found that these artifacts due to SAR-DEM misregistration remain present. We emphasize that the DEM data and ALOS data were collected by different sensors with different spatial resolutions and image geometries. Uncertainties can be further introduced by the filtering techniques applied during image processing. Therefore, it is difficult to fully correct the pixel misregistration issues using the standard InSAR processing software alone. An important contribution of our work is to provide a method for estimating this error at different pixel locations. This enables accurate interpretation of InSAR phase signatures (e.g. the patterns shown in Figure 11 from several real interferograms are DEM artifacts rather than real deformation signals).

*Line 153: could you specify the masking thresholds?*

**Response:** We updated the figure caption of Figure B2 and state that the mask **"excludes any pixels with amplitude dispersion < 0.25 and phase coherence < 0.2 (e.g., water bodies and the area affected by the 2007 Anatuvuk River fire)."** We now add this information in the main text as well.

*Line 163-179: much of its content doesn't fit within the InSAR processing section and could be relocated.*

**Response:** Equation (4) relates the observed seasonal LOS deformation due to the active layer thaw to the amount of water stored in the saturated active layer. We decided to keep Equation (4) and the relevant discussion in Section 2.2, because this equation is built upon Equation (1) and Equation (3), and it shows that 1 cm errors in InSAR LOS deformation measurements can lead to 14 cm error in $z_{water}$ estimates. This naturally leads to Section 2.3, which covers error sources in InSAR LOS deformation measurements.

*Section 3:*
*Figure 5: panel b DEM colorbar's annotations are flipped; please also cite the source of the land cover map in the caption.*

**Response:** Thank you, we corrected the colorbar and added relevant citations in the figure caption.

*Figure 6 caption: What is meant by '4% vector length'?*

**Response:** We edited the last sentence of the figure caption as **"... The normalized Z_water curve was then smoothed using a box car filter with a window size equal to 4% of the number of radar pixels along the transect."**

*Table 1: only need to keep one significant digit after the decimal points, to be consistent with the description in the main text.*

**Response:** We updated Table 1 as suggested.

*I understand the challenges of directly comparing remote sensing estimates with in-situ measurements. I recommend including scatter plots comparing these in the supplementary materials.*
**Response:** As we stated in Section 2.4, a pixel in an InSAR-derived deformation map is ~ 100-by-100 meter, while field measurements were collected at sites with size ~ 900 cm$^2$ (30-by-30 cm plots). The soil layer thickness and the depth to water table measurements can vary substantially within one InSAR pixel at multiple soil pits. This is why the exact point-to-point comparison is not feasible. Instead, we designed a statistical comparison approach to compare the distribution of $Z_{water}$ as inferred by InSAR and field observations under different vegetation covers (Figure 8).

While we devoted substantial amount of effort to collect over 200 soil samples (from these samples, porosity can be measures, we were not able to collect at least three soil samples (one from acrotelm, one from cateletem, and one from mineral soils) at every soil pit due to time constraints and the remote nature of the site. This makes it impossible to generate the scatter plots as suggested by the reviewer.

*Line 308: PALSAR is misspelled.*
**Response:** We have corrected this typo.

*Figure 9: add vertical and horizonal scales for the DEM profile and add a distance scale to panel (c) Add distance scales to Figures 10d, 11a, 12a.*
**Response:** We updated Figures 9, 10, 11, and 12 as suggested.

*Line 428: Since section 3.3 primarily presents and discusses simulations of errors due to 1-2 pixel misregistration in individual interferograms, it is unclear how your approach provides a valuable way to identify and characterize pixel misregistration errors in the final LOS deformation estimates.*
**Response:** In Section 3.3, we showed that the observed phase patterns (Figure 11) closely resemble the patterns of the simulated DEM errors ($\delta$ in Equation 6) due to 1-2 pixel misregistration to east (Figure 12a and Figure 12c). This allows us to conclude that the observed InSAR phase patterns are likely due to DEM errors rather than true deformation signals. We note that the LOS phase error due to DEM-SAR image misregistration is controlled by the amount of pixel misregistration, the local slope, and the InSAR perpendicular baseline. We estimated the InSAR LOS phase error due to 1 pixel misregistration for a perpendicular baseline of 5104 meters in Figure 13. Given that the pixel misregistration error is typically on the order of sub-pixels and the perpendicular baselines in most ALOS interferograms are less than 5104 meters, we reach an important conclusion that InSAR is a feasible technique for regional active layer water storage mapping for a majority of pixels over our study site.

We clarified these points at the end of Section 3.3: "**Nonetheless, our approach provides a method to estimate spatial characteristics and upper bound of InSAR phase errors due to DEM-SAR pixel misregistration in individual interferograms**".

---

## Author Response (AR2)

Dear Editors and reviewers:

    We appreciate the constructive feedback we received, and here we provide detailed explanations of how we revised the paper to address one reviewer's questions. The reviewer comments are in blue, our responses are in black, and relevant text edited or found in the manuscript file is italicized. We thank the reviewers for their insightful comments, which substantially improved the quality of the paper.

**Reviewer 1**
Reviewer 1 did not provide any comments/questions in this round of review.

**Reviewer 2**
**Comment 1 - Line 6:** Maybe consider specifying it is the "equivalent thickness" of soil water that experiences the annual freeze-thaw cycles that ranges from 0 to 75 cm?
**Response:** We followed the reviewer's suggestion and changed Line 6 to: *"... the equivalent thickness of the soil water...".*

**Comment 2 - Line 22-25:** It is not clear to me how the fate of carbon will depend on wetness or dryness. In your analysis, there is no net loss of soil water (line 94: the amount of water that experienced the annual freeze-thaw cycle does not change much – net water drainage ~ 0), no matter whether it is wet or dry. If the water is staying still and just freezing and thawing locally, the link from the soil water storage to the carbon cycle that seems to motivate the entire manuscript from the abstract and the introduction is not apparent to me. I would have thought that soil water is an important topic on its own because it sustains the ecology, and permafrost thaw could bring hydrological or geo – hazards etc.
**Response:** Water saturated soils can deplete oxygen and become anaerobic, which alters the balance of $CO_2$ versus $CH_4$ produced. We clarified this at Line 22-25:
*"Whether the carbon held by... depends largely on the wetness or dryness ... of the active layer, which controls the redox status of the soil that influences the balance of $CO_2$ and $CH_4$ production (Bond-Lamberty et al., 2016; Taylor et al., 2021)".*

*References:*
Bond-Lamberty, B., Smith, A. P., & Bailey, V. (2016). Temperature and moisture effects on greenhouse gas emissions from deep active-layer boreal soils. Biogeosciences, 13(24), 6669–6681. https://doi.org/10.5194/bg-13-6669-2016
Taylor, M. A., Celis, G., Ledman, J. D., Mauritz, M., Natali, S. M., Pegoraro, E. -F., Schädel, C., & Schuur, E. A. (2021). Experimental soil warming and permafrost thaw increase $CH_4$ emissions in an upland tundra ecosystem. Journal of Geophysical Research: Biogeosciences, 126(11). https://doi.org/10.1029/2021jg006376

**Comment 3 - Figure 2:** The newly added cartoon does a good job in illustrating the conceptual model. Column A and column B are rather repetitive and can be combined into one. Column C looks like there is an additional layer of sediment in the frozen layer and overall, which is inaccurate. Could you decide in your model, whether the water will fill in the pore space of a thicker sediment column when frozen, or will expand the pore space of the original saturated sediment column? In either case, the total number of bubble layers should stay constant.
**Response:** We combined columns A and B as suggested, and we edited Column C so that the total number of soil layers remains unchanged in both unfrozen and frozen cases. In our model, water in the unsaturated zone can expand to fill the empty pore space (soil atmosphere) during freezing without contributing to surface deformation. Water in the saturated active layer leads to uplift during freezing.

**NEW FIGURE 2:**

[Figure]

*Figure 2. Conceptual diagram of how the depth of saturated water ($Z_{water}$) affects soil surface deformation. (A) The summer case when the entire active layer is thawed. Here the soil is divided into three zones: (1) the active layer unsaturated zone (thickness of u), which contains soil particles (dark circles) and soil atmosphere (white open space). It may also contain tension (capillary) water (shown in light blue); (2) the water-saturated active layer (thickness of s). The upper surface defines the water table, and the lower surface defines the ice table that separates the thawed active layer from frozen ground (or the permafrost layer at maximum annual thaw depth); and (3) the frozen permafrost (thickness of p). (B) The winter case when the entire active layer is frozen. Water stored in the saturated active layer leads to frost heave ($0.09*Z_{water}$) during freezing. By contrast, water in the unsaturated zone can expand to fill the empty pore space (soil atmosphere) during freezing without contributing to surface deformation (thickness u does not change).*

**Comment 4 - Figure 5:** The land cover map is labelled with 3 land cover classes, but in your data analysis in Table 1 and Figure 7, you added a 4th "Wet Tussock" class. In Figure B4, you show three field photos including a wet sedge instead of wet Tussock. It would be nice to be more consistent. Could you also label Fig 7 against the red and purple polygons in panel c, as it is not described in the caption, only mentioned in the capture of Figure 7.

**Response:** In this study, we mainly focused on three land covers: "Heath", "Tussock", and "Sedge". We found that the active layer tends to be wetter as the terrain becomes flatter in the northern part of the study area (Path 255 Frame 1380). Thus, we refer to the "tussocks" pixels in regions highlighted in Figure 7 (d)-(f) as "wet tussock".

We clarified at Line 316-320: "*Figure 7 (d)-(f) shows another zoomed-in area from frame 1380 (with the location outlined in a red dashed line in Figure 5(c)), where the terrain transitions from rolling hills to coastal plains. Here, tussocks are the dominant land-cover type, and water-loving shrubs and sedges are distributed along the water tracks (visible in the optical image). Because this region is wetter than the Toolik Lake area, we refer to the tussock pixels in Figure 7 (d)-(f) as "wet tussock" in Figure 7 (g) and Table 1*". We also changed the caption of Figure B4 to "*... (c) sedge land cover...*".

In Figure 7, we added purple outlines to panel (a)-(c) and red outlines to panel (d)-(f) to better illustrate the image locations in Figure 5. We have changed the red solid and dashed outlines to orange to avoid confusion and updated the figure caption accordingly.

**NEW FIGURE 7:**

[Figure]

**Comment 5 - Figure 14:** The caption says the locations of these pixels are shown in Figure 10. I see one location of Pw and one location of Pe and if we are looking at all interferograms with phase difference extracted from these two locations and from the Pf pixel, why should there be a range of slopes?
**Response:** As shown in Figure 10-11, InSAR phase measurements around point $P_W$ and $P_E$ are noisy, and the slope shows relatively large spatial variations. To generate Figure 14, we calculated the average phase difference across a ridge line ($P_E$-$P_W$) and across a flat area (around point $P_f$) to estimate the LOS errors at these two locations. Here phase averaging around a point of interest serves as a spatial box filter, which reduces random phase noises associated with surface vegetation.
We rephrased the caption of Figure 14 to: *"... perpendicular baseline (in m) at $P_E$-$P_W$ with an average percent slope of 10.4% (blue dots) and around $P_f$ with an average percent slope of 1.9% (orange dots)"*.
In this updated caption, we reported the average slope of all pixels used for estimating the LOS errors..

**Comment 6 - Line 449:** In various places in text, you concluded that 95% of $Z_{water}$ estimates range from 0 to 62 cm (and I assume the whole range is 0-75 cm). This can only result from a subsidence map that is strictly negative, as shown in Figure 5a. I wonder whether there is any positive value detected in the InSAR thaw subsidence map due to the misregistration of DEM pixels, given how the offsets should produce both positive and negative errors rather symmetrically. If not, is it a lucky event? Or is it expected as it is the valleys that are wetter and subside faster and experience more of an error with pixel corregistration? Or maybe could we have a $Z_{water}$ map against an estimated error map to show the uncertainties in map new? Figure 5 and Figure 13 currently do not have the same extent.
**Response:** As shown in Figure 5(a), InSAR suggested a majority of pixels experienced centimeter-level thaw subsidence between early June and late July. There are positive values below the noise level (<1cm) detected at a small number of InSAR pixels, mostly around the reference point (a dry highland area with relatively flat terrain; marked in Figure B1) where the minimum seasonal deformation is expected. The positive values were due to noise (e.g., residual decorrelation noise and residual tropospheric noise) added to ~ 0 cm of deformation signal. We also want to note that up to 75 cm $Z_{water}$ was observed in the wettest riparian zone after removing less than 3% of outliers. We clarified this in the first paragraph of Section 3.1.
We have updated Figure 13 to cover the same area as Figure 5, and our conclusion remains the same: the pixel misregistration artifacts are mostly observed in a small subset of pixels with relatively large slope angles (located in the southernmost portion of the study area). As shown in Figure 10, while

we observed a phase difference across the ridge line ($P_E$-$P_W$) in early June to late July interferograms, the LOS phase observations are ≥ 0 radians on both east and west sides of the ridge (Here pink indicates positive LOS or subsidence and blue indicates minimum deformation) with respect to the reference point (marked in Figure B1). We also presented additional interferograms in the zoom-in region in Figure 11. As noted in Figure 11 caption, all interferograms here were referenced to a local reference point near $P_E$ (not the reference point marked in Figure B1). This is why, we see both positive and negative phases in Figure 11.

**NEW FIGURE 13:**

[Figure]

In addition, could we have some statistics on the subsidence magnitude (or $Z_{water}$) vs slope (scatter or box-and-whisker) to understand whether the 14 cm of error are associated with the high-end of the $Z_{water}$ values or not?

**Response:** As we discussed in Section 2.3. the <1cm error in LOS measurement (equivalent to <14 cm error in $Z_{water}$) could be due to residual tropospheric turbulence noise. Topographic artifacts related to DEM-SAR pixel misregistration are most noticeable in areas with a slope larger than 10% (Figure 13). We emphasize that the DEM error δ is determined by both the slope angle and the amount of pixel misregistration. When there is no pixel misregistration, DEM error equals 0 regardless of the slope angles. Furthermore, while topography influences soil water distribution ($Z_{water}$), there are many other factors that

can play a role. Therefore, we cannot identify any clear relationship between subsidence magnitude  vs slope in a scatter plot.